# Sentinel-1 Detection of Ice Slabs on the Greenland Ice Sheet

Riley Culberg[1], Roger J. Michaelides[2], and Julie Z. Miller[3]

[1]Department of Earth and Atmospheric Sciences, Cornell University
[2]Department of Earth, Environmental, and Planetary Sciences, Washington University in St. Louis
[3]Earth Science and Observation Center, Cooperative Institute for Research in Environmental Sciences, University of Colorado Boulder

**Correspondence:** Riley Culberg (rtculberg@cornell.edu)

**Abstract.** Ice slabs are multi-meter thick layers of refrozen ice that limit meltwater storage in firn, leading to enhanced surface runoff and ice sheet mass loss. To date, ice slabs have primarily been mapped using airborne ice-penetrating radar, which has limited spatial and temporal coverage. This makes it difficult to fully assess the current extent and continuity of ice slabs or to validate predictive models of ice slab evolution that are key to understanding their impact on Greenland's surface mass balance. Here, for the first time, we map the extent of ice slabs and superimposed ice facies across the entire Greenland Ice Sheet at 500 m resolution using dual-polarization Sentinel-1 (S-1) synthetic aperture radar data collected in winter 2016-2017. We do this by selecting empirical thresholds for the cross-polarized backscatter ratio (HV - HH) and HV backscattered power that jointly optimize agreement between airborne ice-penetrating radar data detections of ice slabs and the S-1 estimates of ice slab extent. Our results show that there is a strong correlation between C-band backscatter and the ice content of the upper $\sim$ 7 meters of the firn column that enables ice slab mapping with S-1. Our new mapping shows that ice slabs are nearly continuous around the entire margin of the ice sheet. This includes regions in Southwest Greenland where ice slabs have not been previously identified by ice-penetrating radar, but where the S-1 inferred ice slab extent shows strong agreement with the extent of visible runoff mapped from optical imagery. The algorithm developed here lays the groundwork for long-term monitoring of ice slab expansion with current and future C-band satellite systems and highlights the potential added value of future L-band missions for near-surface studies in Greenland.

## 1 Introduction

Over the last two decades, around half of mass loss from the Greenland Ice Sheet (GrIS) has come from the runoff of surface meltwater , with the remaining 45-50% attributable to ice dynamical processes and ice-ocean interactions in marine terminating sectors (Van Den Broeke et al., 2009; Enderlin et al., 2014; Mouginot et al., 2019). Surface processes are projected to remain the dominant contributor to Greenland's sea level contribution over the next century, particularly as the ice margin retreats onto land above sea level (Fox-Kemper et al., 2021). By extension, much of the uncertainty in future mass loss from the ice sheet can also be ascribed to uncertainty in surface processes (Fox-Kemper et al., 2021). One such process that remains poorly constrained is the development and expansion of ice slabs in firn, particularly near the equilibrium line. Ice slabs are multi-meter thick layers of refrozen ice that form just below the surface (Machguth et al., 2016) and can be horizontally

continuous over tens of kilometers (MacFerrin et al., 2019). As a result, ice slabs are largely impermeable and limit the vertical percolation of meltwater into the underlying relict firn, leading to a rapid transition from meltwater retention to runoff as they form (Machguth et al., 2016; MacFerrin et al., 2019; Tedstone and Machguth, 2022). To date, ice slabs have primarily been mapped using Operation IceBridge (OIB) airborne ice-penetrating radar surveys, as these data directly resolve the vertical structure of the subsurface and can distinguish homogeneous refrozen ice bodies from lower density firn (MacFerrin et al., 2019; Jullien et al., 2023). These data have shown that ice slabs dominate the wet snow zone along the western, northern, and northeastern coasts of Greenland. The southeast basin is the only major region where limited ice slabs have been detected, due to the high snow accumulation rate that insulates subsurface liquid water from refreezing and leads to the formation of perennial firn aquifers instead (Forster et al., 2014; Munneke et al., 2014).

While the OIB data have provided critical insights into ice slab extent across the GrIS, these data are significantly limited in both space and time. Data are only available directly beneath the aircraft flight track, and collection was limited to a moderate number of flight lines in spring (typically April or May) each year from 2011-2014 and 2017-2018, along with a few additional flights over the wet snow zone in 2010. These gaps in coverage lead to a number of issues. In many regions, the upper elevation limit of the ice slabs is poorly defined, due to a lack of flights perpendicular to the coastline, and there are some areas, most notably in southern Greenland and on peripheral ice caps, where there is insufficient flight coverage to assess whether ice slabs are present at all. Even in regions of good coverage, there are typically 5-20 km gaps between flight lines. As a result, the full extent of ice slabs on the GrIS remains poorly defined and it has been difficult to assess the km-scale continuity of this facie. Additionally, there are very few repeated flights that were flown perpendicular to the coastline, which are required to robustly assess the inland expansion of ice slabs from year to year. Jullien et al. (2023) showed that some ice slab growth occurred between the period from 2010-2012 to 2017-2018, but the resolution and coverage of that analysis was limited by large spatial data gaps and the need to aggregate multiple years of data to achieve reasonable coverage of the whole ice sheet. With the end of the OIB mission in 2019, there are no current or planned ice-penetrating radar missions to improve these time series or to assess the impact of more recent heavy melt seasons, such as 2019, 2021, and 2023, which included a significant high elevation rain event in August 2021 (Tedesco and Fettweis, 2020; Box et al., 2022, 2023).

These spatial and temporal gaps significantly impede our ability to assess the impact of ice slab development and expansion on the current and future mass balance of the GrIS. For example, MacFerrin et al. (2019) parameterized ice slab extent as a function of the ten-year running mean of local excess melt and applied this parameterization to an ensemble of regional climate models to predict that ice slab expansion would add 7-74 mm of additional sea level rise by 2100. However, this excess melt threshold was tuned by matching the modeled ice slab extent to the aggregate observed extent from 2010-2014 (MacFerrin et al., 2019). As a result, it remains unclear whether the temporal evolution of ice slabs in this model accurately captures the true pace of ice slab growth. Recently, more physics-based models of firn hydrology have been used to model climatic drivers of ice slab extent and expansion (Brils et al., 2024), but in the absence of validating data, significant uncertainties in future projections will remain.

The only clear mechanism for mapping ice slab extent across the entire ice sheet at high resolution (∼1 km or better) on an annual or better basis is to use satellite microwave remote sensing systems. In fact, ice slabs have been mapped from space

using the L-band radiometer onboard the Soil Moisture Active Passive (SMAP) mission in Miller et al. (2022a, b). However, there are limitations to that algorithm. In particular, the instrument resolution is approximately 30 km (Miller et al., 2022a), making it difficult to clearly define the inland extent of the ice slabs and impossible to capture expansion on the order of a few kilometers or less per a year. Additionally, although rough estimates of the interannual variability are given, this algorithm aggregates ∼5 years of radiometer data to create a single estimate of ice slab extent to create higher probability maps (Miller et al., 2022a), which limits its use for generating long time series. There are also notable discrepancies between the SMAP and OIB ice slab extents, particularly in the Northwest where SMAP fails to detect large swaths of the OIB-detected ice slabs, and in the North and Northeast where SMAP places the ice slabs at higher elevations than the OIB data (see Fig. 11).

An alternate approach is to use active synthetic aperture radar systems such as the ESA Sentinel-1 (S-1) series satellites (Berger et al., 2012). Since C-band radio waves penetrate roughly 5-15 meters into snow, firn, and ice, depending on the local physical and dielectric properties (Rignot et al., 2001; Hoen, 2001; Fischer et al., 2019), the depth-integrated surface echo measured by the instrument mainly contains information about the near-surface structure. In Extra Wide Swath mode, Sentinel-1 covers the entire GrIS approximately every 10 days with a spatial resolution of 20 x 40 meters and data are available from late 2014 to the present day. With the anticipated launches of Sentinel-1C & D, the data record is projected to continue uninterrupted through at least the early 2030s. Therefore, Sentinel-1 could not only provide the first pan-Greenland mapping of ice slabs at high-resolution, but such an algorithm would open the door to long-term monitoring of ice slab expansion, potentially covering close to two decades of observations. Here, we develop an algorithm to map refrozen ice facies on the Greenland Ice Sheet using dual-polarization Extra Wide Swath Sentinel-1 measurements of radar backscatter in conjunction with calibration data from ice-penetrating radar observations.

## 2  Electromagnetic Interactions in Firn

On ice sheets, mean firn density increases exponentially with depth as it compacts under its own weight (Bader, 1954; Herron and Langway, 1980). In the percolation zone, the structure is further modified by the infiltration and refreezing of surface meltwater that forms ice lenses and ice pipes (Benson, 1962). Ice lenses are horizontal sheets of refrozen solid ice that may be up to a few tens of centimeters thick and extend laterally for a few meters (Benson, 1962; MacFerrin et al., 2019), while ice pipes are vertical refrozen conduits that represent preferential infiltration pathways connecting these ice lenses (Marsh and Woo, 1984; Pfeffer and Humphrey, 1998; Humphrey et al., 2012). The proportion of the firn column occupied by these refreeze features generally increases with decreasing elevation and increasing melt-to-accumulation ratio (Harper et al., 2012; Machguth et al., 2016). In the extreme, consistent excess melting may anneal these ice lenses together into multi-meter thick ice slabs that form in the wet snow zone (MacFerrin et al., 2019; Machguth et al., 2016). The wet snow facies eventually transition to the ablation zone via a region of superimposed ice facies, where the near-surface ice is formed by refreezing within the annual accumulation (Benson, 1962). At the lowest elevations, where annual melting consistently exceeds accumulation, the ice sheet transitions to the bare ice ablation zone composed of homogeneous meteoric ice that is exposed at the surface via horizontal advection and ablation.

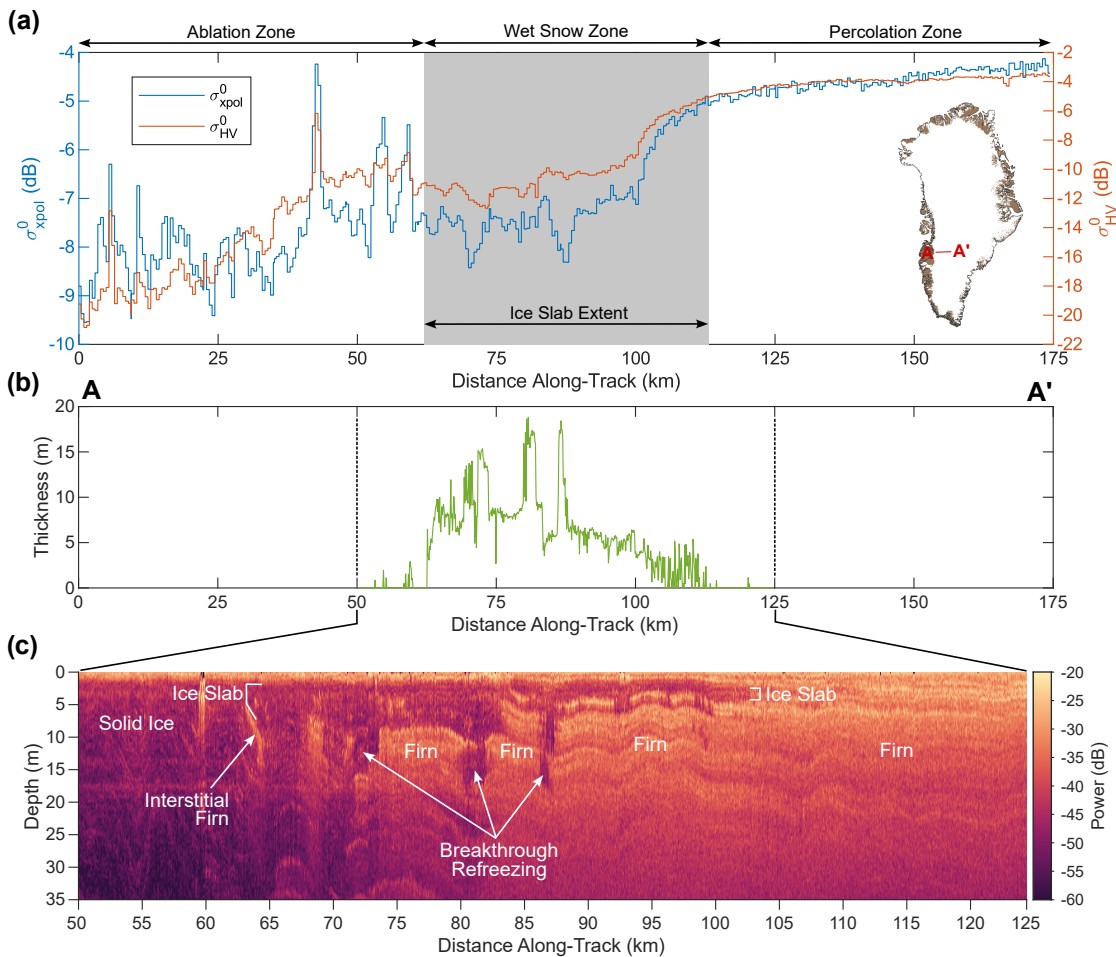

**Figure 1.** Radar signatures of ice slabs along a transect in Southwest Greenland. a) Sentinel-1 $\sigma^0_{HV}$ is shown in red and the cross-pol backscatter ratio, $\sigma^0_{xpol} = \sigma^0_{HV} - \sigma^0_{HH}$ is shown in blue. The gray region denotes where ice slabs have been detected with ice-penetrating radar (Culberg et al., 2022b). The inset map (Gerrish, 2020; Morlighem et al., 2017) shows the location of this transect in Southwest Greenland. b) Ice slab thickness along the transect as measured with ice-penetrating radar (Culberg et al., 2022b). There is a rapid, down-flow decrease in $\sigma^0_{xpol}$ as the ice slab thickens, with the backscatter plateauing once the ice slab reaches a thickness of around 7 m. c) Radargram from April 2015 collected by the Ultrawideband MCoRDS system (Paden et al., 2014a) showing the subsurface structure in the region where ice slabs have been detected. In the percolation zone, the structure is dominated by layered firn with strong scattering from embedded ice features. In the wet snow zone, a thick layer of homogeneous refrozen ice with low backscatter overlies relict firn. In the ablation zone, only solid ice remains and there is relatively low backscatter at all depths due to the absence of density contrasts in the subsurface. (Note: This figure contains modified Copernicus Sentinel data 2016-2017, processed by ESA.)

These near-surface structural variations with elevation lead to commensurate changes in the dominant electromagnetic scattering mechanisms. In the percolation zone, radar echoes are thought to be dominated by volume scattering from embedded

ice features on the scale of a few wavelengths (Fahnestock et al., 1993; Jezek et al., 1994; Rignot, 1995; Baumgartner et al., 1999; Langley et al., 2009), making the GrIS percolation zone one of the most radio bright regions on Earth (Swift et al., 1985; Rignot et al., 1993; Jezek et al., 1994). Past work has modeled the observed percolation zone backscatter at C-band as volume scattering from randomly oriented cylinders (Rignot, 1995). This volume scattering dominated regime also leads to significant depolarization of the incident wave and a large radar cross-section in the cross-polarized (HV or VH) channels (Jezek et al., 1993; Rignot, 1995; Langley et al., 2007; Barzycka et al., 2019). By contrast, scattering in the bare ice ablation zone is dominated by rough surface scattering at the air-ice interface, with relatively little volume scattering since heterogeneities such as air bubbles are significantly smaller than the C-band wavelength (Langley et al., 2007, 2009; Barzycka et al., 2019). As a result, the radar cross section of the ablation zone is relatively low and little depolarization occurs, so the echoes are dominated by co-polarized (HH or VV) returns (Langley et al., 2007, 2009). Numerous papers have mapped glacier facies on Arctic ice caps and mountain glaciers based on these characteristic changes in backscatter (Partington, 1998; Long and Drinkwater, 1994; Barzycka et al., 2019). For example, Langley et al. (2008) demonstrated that on Kongsvegen Glacier in Svalbard, the boundaries between firn, superimposed ice, and glacier ice could be mapped in C-Band ENVISAT SAR data from the $\sim$5 dB change in backscatter between each region, with ground-penetrating radar used to validate the mapping.

Ice slab regions likely represent an intermediate scattering regime between the percolation zone and superimposed ice or ablation zones, with both surface and volume scattering contributing to the total backcatter. Nadir-looking airborne radar sounding measurements show that ice slabs are characterized by strong reflections from their upper and lower interfaces, but very low backscatter within the refrozen ice itself (MacFerrin et al., 2019; Jullien et al., 2023). However, the presence of remnant interstitial firn layers does lead to overall higher radar sounder backscatter in these refrozen ice facies than in meteoric ice (Fig. 1c). In synthetic aperture radar (SAR) returns, the ratio of the HV to HH backscatter ($\sigma^0_{xpol} = \sigma^0_{HV} - \sigma^0_{HH}$ (in dB)), known at the cross-polarized backscatter ratio (Ulaby and Long, 2014) or linear backscatter ratio (Rignot, 1995), has been used as a proxy for the ratio of volume to surface scattering in the Greenland percolation zone (Rignot, 1995) and is also responsive to this change in subsurface structure. SAR returns from ice slabs display lower $\sigma^0_{xpol}$ than the percolation zone, but higher values than the upper ablation zone, which could be interpreted as suggesting greater surface scattering and lower volume scattering in ice slab areas compared to the percolation zone, but higher volume scattering than meteoric ice in the upper ablation zone.

Figure 1a-b shows an example of this effect along a transect from the ice margin up to the shallow percolation zone in Southwest Greenland. The percolation zone HV backscatter ($\sigma^0_{HV}$) is consistently about -4 dB, but decays at lower elevations as ice slabs begin to form and thicken, eventually plateauing around an average of -11 dB across the upper ablation and wet snow zones. However, there is significant local variability in these regions, with $\sigma^0_{HV}$ varying from -13 dB to -6 dB around the mean. The cross-polarized backscatter ratio ($\sigma^0_{xpol}$) similarly decreases from -4.5 dB to -7.5 dB as ice slabs develop at this test site. In this paper, we exploit this apparent reduction in volume scattering that occurs over ice slab areas to map ice slabs from S-1 C-band winter backscatter measurements.

## 3 Methods

### 3.1 Sentinel-1 Backscatter Mosaics

For this analysis, we use Extra Wide Swath (EW) ground range detected (GRD) Sentinel-1A & B data collected in HH and HV polarizations at a center frequency of 5.405 GHz (ESA, 2023) over the GrIS from 01 October 2016 to 30 April 2017. We focus on a single year of measurements to demonstrate the feasibility of mapping ice slabs with S-1 data without confounding complications from instrument radiometric stability, evolving observation strategies, and multi-annual changes in surface scattering properties. Only ∼10 days of data are needed to fully cover the entire ice sheet, but we choose to use the full winter

period because the extra observations allow us to develop a robust mean backscatter map that reduces the influence of temporal variability in scattering properties, speckle, and variable incidence angles. We expect ice slab extent to be stable during this period since there is no melt infiltration. We only use winter data because the presence of surface meltwater increases both the surface dielectric contrast and the near-surface attenuation in water-saturated layers, obscuring the subsurface structure. Due to the huge data volume, we process these data in Google Earth Engine (GEE) (Gorelick et al., 2017). Data in the GEE S-1 GRD

data collection have undergone thermal noise removal, radiometric calibration, geometric terrain correction, and conversion to dB values in the Sentinel-1 Toolbox before being posted to the cloud. Unfortunately, these data have not undergone radiometric terrain correction, and it is impossible to fully implement this algorithm in GEE since it requires access to the data in the original radar coordinates. We experimented with applying an angle-based radiometric terrain correction method designed for GEE (Vollrath et al., 2020), but found that it produced little to no change in the backscatter values due to the extremely low surface slopes over most of the ice sheet. Therefore, we do not implement this correction in our final workflow.

With both Sentinel-1A and Sentinel-1B in operation, the exact repeat interval for any point on the ice sheet is 6 days. However, because the EW swath width is 410 km and Greenland is at high latitudes, the coverage is often more frequent. During our 7 month study period, the average number of observations per pixel was 190, or almost one observation per day, with a minimum of 29 and a maximum of 571 observations. Within each observing pass, the incidence angle varies from $18.9°$ to

$47°$ across the swath (ESA, 2023). Particularly in the percolation zone, backscatter varies strongly with incidence angle, which leads to obvious seams between overlapping swaths and spatial variations in backscatter that are attributable to observation geometry rather than physical properties of the ice sheet. Additionally, the data are subject to speckle and temporal variations in backscatter due to snowfall, wind scour, and other environmental factors that impact the surface scattering structure. All of these factors lead to significant challenges in generating a single consistent backscatter mosaic for the entire ice sheet.

To reduce speckle, we first multilook all images to 500 m resolution, which effectively balances speckle suppression and data resolution (see Supplementary Information, Section 1 for the resolution sensitivity analysis). Following prior studies with C- and L-band satellite radar scatterometery data over ice sheets, we correct for incidence angle variations in space and time by fitting a linear function to incidence angle vs. backscatter on a per-pixel basis using all available images in our study period(Long and Drinkwater, 1994; Ashcraft and Long, 2005; Lindsley and Long, 2016; Long and Miller, 2023; Lindsley and

Long, 2016; Long and Miller, 2023). We then use these linear relations to calculate the theoretical backscatter at an incidence angle of $35°$ in each pixel. Radar scatterometry studies have typically corrected their data to an incidence angle of $40°$, but

here we choose to correct the data to an incidence angle close to the middle of the S-1 scene. We combine both ascending and descending orbits from both satellites to maximize the angular diversity in each pixel for the most robust fit and calculate a separate linear fit for the $\sigma_{HH}^0$ and $\sigma_{HV}^0$ measurements.

Supplementary Information, Section 2 provides detailed information on the backscatter residuals after correction and the sensitivity of the final results to the angular diversity and number observations per pixel. Overall, we find that the residuals are generally less than $\pm$ 1 dB and most large residuals are driven by non-stationary backscatter time series in regions with subsurface meltwater, like firn aquifers or subsurface lakes, rather than a failure of the linear fit. However, we also observe large residuals in peripheral regions of the ice sheet and on small ice caps with steep terrain. In general, we find that the final

ice slab classification results are insensitive to per-pixel angular diversity or the number of observations as long as there are a median of at least $\sim$117 observations per pixel spanning at least 10 unique incidence angles. However, if the mean incidence angle of the observations meets or exceeds $30°$, then these thresholds can be reduced to 77 observations per pixel or an angular diversity of $7°$ or greater, a set of criteria which is met for the portion of our study area that covers the ice slabs (see Fig. C4). A small portion of the ice sheet interior in the dry snow zone does not meet the number of observations threshold, but the entire

ice sheet meets the angular diversity threshold.

Our linear fit method not only removes backscatter variations due to observing geometry, but also serves to average all available observations in each pixel. This further reduces speckle and averages out temporal variations in backscatter over the winter season. In this way, we form consistent mean winter backscatter mosaics for the entire ice sheet for each polarization. We then calculate the $\sigma_{xpol}^0$ map by subtracting the $\sigma_{HH}^0$ map from the $\sigma_{HV}^0$ map. Finally, we use the BedMachinev4 ice mask

to remove pixels in regions without ice (Morlighem et al., 2017). Figure 2a-b shows the mean winter $\sigma_{HV}^0$ and $\sigma_{xpol}^0$ mosaics for Greenland in winter 2016-2017. Regions with ice slabs clearly show greater $\sigma_{HV}^0$ than the lower ablation zone, but reduced $\sigma_{HV}^0$ compared with the percolation zone. Similarly, ice slabs show lower $\sigma_{xpol}^0$ values than the percolation zone.

### 3.2   Excluding the Dry Snow Zone and Firn Aquifer Regions

In order to reduce false positive detection of ice slabs, we exclude regions of the ice sheet that a) experience little to no melting

or b) are already determined in previous studies to host firn aquifers. This step is critical because, as can be seen in Fig. 2b, both of these regions exhibit low $\sigma_{xpol}^0$ values that are on par with what is observed in known ice slab regions. In the dry snow zone, this occurs because the subsurface is dominated by smooth depositional snow layers with little heterogeneity beyond the ice grain scale. Firn aquifer regions retain liquid meltwater through the winter which leads to increased subsurface absorption and therefore a relatively greater degree of surface scattering, since subsurface volume scattering is suppressed (Brangers et al.,

2020; Miller et al., 2022a).

To exclude regions with minimal surface melting, we adapt an existing method for mapping wet snow facies in Greenland based on the change in S-1 $\sigma_{HH}^0$ between winter and summer (Hu et al., 2022). Much like the classic radar scatterometer and microwave radiometer algorithms for mapping surface melting and firn saturation from VV backscatter (Wismann, 2000; Ashcraft and Long, 2006; Hicks and Long, 2011; Miller et al., 2022b), this approach exploits the fact that the enhanced

microwave absorption in wet snow leads to a significant reduction in backscatter during the summer when surface melting

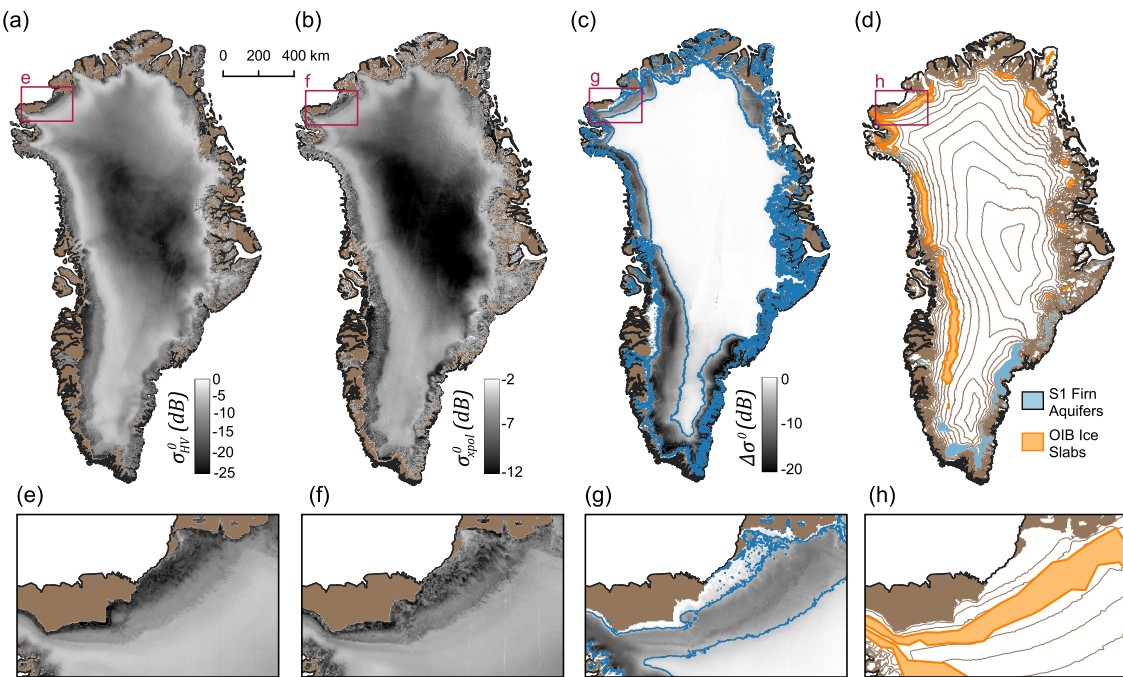

**Figure 2.** a) Average winter $\sigma^0_{HV}$ map at $35^o$ incidence angle covering 1 Oct 2016 - 30 April 2017. b) Average winter $\sigma^0_{xpol}$ map at $35^o$ incidence angle covering 1 Oct 2016 - 30 April 2017. c) Difference between summer and winter HH backscatter ($\Delta\sigma^0$), averaged over 1 Nov 2014 - 31 Aug 2020 . We only consider the regions inside the blue outline for our ice slab analysis, since the large change in backscatter between seasons (dark color) indicates that there is significant annual surface melt retained in surface snow in these areas. d) Locations of firn aquifers (blue) detected using Sentinel-1 data from 2014-2019 as published in Brangers et al. (2020). Regions detected as ice slabs with ice-penetrating radar data (Jullien, 2023) are shown in orange for reference. For each mosaic, the zoom-in panels show details of the North Greenland ice slab region. e) North Greenland $\sigma^0_{HV}$. f) North Greenland $\sigma^0_{xpol}$. g) North Greenland $\Delta\sigma^0$. h) North Greenland OIB ice slab detections. In all panels, the Greenland coastline was produced by the British Antarctic Survey (Gerrish, 2020), the ice mask as part of BedMachinev4 (Morlighem et al., 2017), and the 200 m contours are derived from ArcticDEM (Porter et al., 2018). (Note: Panels a-c contain modified Copernicus Sentinel data 2016-2017, originally processed by ESA.)

occurs. We first create an average winter $\sigma^0_{HH}$ map at 35° incidence angle by applying the same linear correction method described in Section 3.1 to aggregated data from 1 Nov - 31 March each year between 2014 and 2020. We then create an average summer $\sigma^0_{HH}$ map using all observations between 1 July and 31 Aug from 2015-2020, corrected to 35° in the same manner. Finally, we calculate the difference between the summer and winter backscatter as $\Delta\sigma^0 = \sigma^0_{summer} - \sigma^0_{winter}$. We aggregate data over these five years because melt extent varies significantly from year to year and from region to region. This extended time series prevents us from inadvertently excluding areas from analysis due to anomalously low melt extent in any given year, despite a sufficient history of melt to have formed ice slabs. Additionally, it ensures we have sufficient observations with sufficient angular diversity during the three month summer period. We then choose an empirical threshold to discriminate

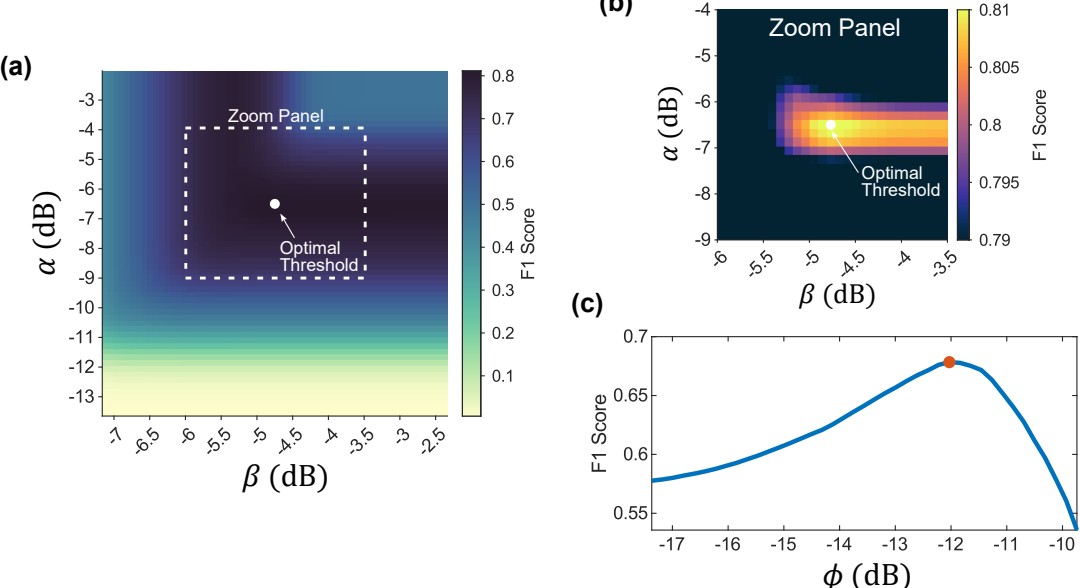

**Figure 3.** Selection of the optimal thresholds for ice slab detection. a) F1 score for delineating the upper elevation limit of the ice slabs as a function of $\alpha$ and $\beta$ thresholds. The optimal threshold combination (maximum F1 score) is shown in the white dot. b) Zoom-in of the region around the optimal threshold combination, showing the global maximum in F1 score. c) F1 score for delineating the lower elevation limit of the ice slabs as a function of $\phi$ threshold. The optimal threshold (maximum F1 score) is shown in the red dot.

regions with consistent surface melting. Hu et al. (2022) derived a threshold of -7 dB to discriminate between wet snow facies
and the percolation zone in $\Delta\sigma^0$ images, based on the distribution of backscatter values observed in Northeast Greenland. However, we find that this threshold is overly aggressive when applied to our average $\Delta\sigma^0$ map and excludes some regions in North Greenland where ice slabs have been observed with ice-penetrating radar. Therefore, we use a threshold of $\Delta\sigma^0 <$ -4.7 dB, which is the minimum value that produces a melt region mask which encompasses all OIB ice slab observations from spring 2017. This threshold value falls midway between the Hu et al. (2022) threshold of -7 dB for discriminating wet snow
facies and the common threshold of -3 dB for discriminating regions of surface melting (Nagler and Rott, 2000; Liang et al., 2021; Li et al., 2023), suggesting that this is a reasonable empirical choice that is consistent with prior work on wet snow mapping with S-1. Figure 2c shows the five-year melt extent mosaic, with the region we consider for ice slab detection ($\Delta\sigma^0$ < -4.7 dB) outlined in blue.

To exclude firn aquifer regions, we use the Sentinel-1 firn aquifer map originally published in Brangers et al. (2020). These
firn aquifer areas were detected by identifying pixels where the mean April $\sigma^0_{HV}$ exceeded the mean September $\sigma^0_{HV}$ by 9.4 dB or more, using mean monthly values aggregated over 2014-2019, similar to our melt area map. Figure 2d shows the locations of these firn aquifers in relation to previous OIB ice slab detections.

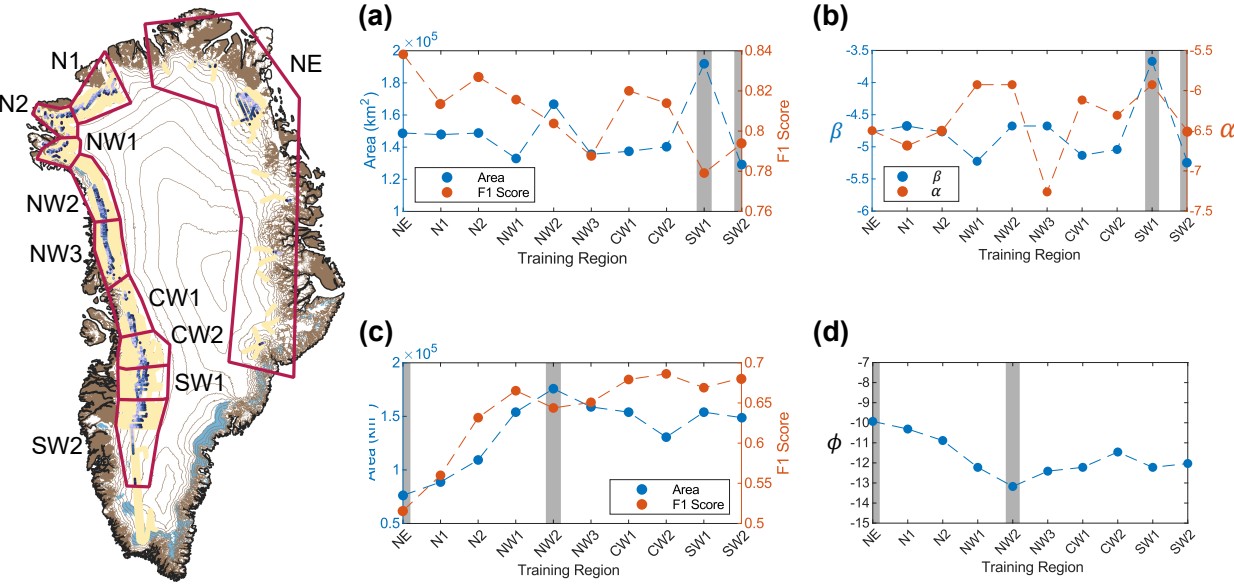

**Figure 4.** Results of the 10-fold cross-validation scheme. The overview map of Greenland shows the ten training regions. In each panel, the training regions that produce the maximum and minimum total ice slab extent are marked in the grey bars. a) The total ice slab area and F1 score on the withheld validation set for each iteration of the ten-fold cross-validation of the ice slab upper elevation limit. b) The estimated values of $\alpha$ and $\beta$ derived from each of the ten training regions. c) The total ice slab area and F1 score on the withheld validation set for each iteration of the ten-fold cross-validation of the ice slab lower elevation limit. d) The estimated values of $\phi$ derived from each of the ten training regions.

### 3.3 Threshold Optimization and Uncertainty Analysis

Sections 3.1 and 3.2 describe how we form $\sigma^0_{xpol}$ and $\sigma^0_{HV}$ mosaics over high-melt regions where ice slab formation might
be possible. To then map ice slab extent, we choose backscatter thresholds that can delineate regions with ice slabs from regions without ice slabs. We assess uncertainty by quantifying the range of plausible S-1 inferred ice slab extents that would be consistent with the OIB airborne ice-penetrating radar observations. We approach this problem in two steps. First, we use all available OIB ice slab detections to find the optimal backscatter thresholds that produce the best ice-sheet-wide agreement between the S-1 inferred ice slab extent and the OIB ice slab extent. By applying these optimal thresholds to the backscatter
mosaics, we produce a map of the most likely ice slab extent across the ice sheet. Then, to assess uncertainty, we use a 10-fold cross validation scheme where we generate 10 new sets of thresholds, each optimized using only a small subset of the OIB data. From the results of these ten trials, we use the backscatter thresholds that produce the largest total ice slab area to define the maximum plausible ice slab extent, and the thresholds that produce smallest total ice slab area to define the minimum plausible ice slab extent. Together, this quantifies the range of plausible S-1 inferred ice slab extents that are still an acceptable fit to the
OIB observations. Below, we describe in detail how we choose these thresholds.

### 3.3.1 Most Likely Ice Slab Extent

We use a training data set built from the Jullien et al. (2023) high-end estimate of ice slab extent derived from OIB flight lines surveyed in March-May 2017. (This high-end estimate corresponds to the maximum likely refrozen ice content given the observed ice-penetrating radar signal strength.) For each flight line that passes through an ice slab area, we extract the portion of the flight line that overflies the ice slabs, as well as an additional ∼50 km buffer that extends inland of the upper limit of the ice slabs. We discretize these lines into points every 50 m and assign each point a value of 1 if an ice slab was detected in the OIB data at that location or 0 if no ice slab was detected. These observations are then used to optimize the backscatter thresholds. We use a brute force search to find optimal values of $\alpha$ and $\beta$ that maximize the agreement between the upper elevation limit of the ice slabs as detected by airborne ice-penetrating radar, and the upper limit of the ice slabs as estimated by S-1. Areas where $\sigma^0_{HV} < \alpha$ and $\sigma^0_{xpol} < \beta$ are taken to be ice slabs. We test all combinations of thresholds where -13.6 dB< $\alpha$ <-2.1 dB and -7.12 dB < $\beta$ < -2.37 dB, calculate the F1 score for each combination, and choose the threshold values that give the highest F1 score. The F1 score is a measure of the accuracy of a binary classification and is calculated following Equation 1.

$$\text{F1} = \frac{2 * \text{true positive}}{2 * \text{true positive} + \text{false positive} + \text{false negative}} \tag{1}$$

Figure 3 shows this optimization trade space and the optimal values of $\alpha$, $\beta$, and $\phi$ that we derive. We find that using both $\sigma^0_{xpol}$ and $\sigma^0_{HV}$ thresholds together leads to modestly better agreement with the OIB detections, compared to using only $\sigma^0_{xpol}$. When only $\sigma^0_{xpol}$ is used to delineate the upper elevation limit of the ice slabs, the maximum F1 score is 0.787, compared to a maximum F1 score of 0.811 when both backscatter thresholds are used. When using only $\sigma^0_{HV}$ to delineate the upper elevation limit of the ice slabs, the maximum F1 score is only 0.674, so it is clear that $\sigma^0_{xpol}$ provides additional information that improves the delineation of the upper boundary.

Initial analysis of the backscatter mosaics suggests that $\sigma^0_{xpol}$ does not display an unique change in behavior associated with the lower boundary (see Fig. 2), so we optimize a separate threshold, $\sigma^0_{HV} > \phi$, to delineate the lower elevation limit of the ice slabs. We optimize $\phi$ following the same method as described above, but using a new version of the OIB training dataset that covers the ice slab region and a ∼50 km buffer down-flow into the ablation zone. Altogether, the area defined by $\sigma^0_{xpol} < \beta$ and $\phi < \sigma^0_{HV} < \alpha$ is our most likely estimate of the spatial extent of ice slabs across the ice sheet.

### 3.3.2 Maximum and Minimum Ice Slab Extent

To quantify uncertainty in this most likely estimate of ice slab extent, we use a 10-fold cross validation scheme. We divide our training dataset into 10 subsets, each containing OIB ice slab detections from a different region of the ice sheet (see Fig. 4). For each of the ten regions, we again use a brute force search to find the values of $\alpha$, $\beta$, and $\phi$ that produce the best agreement between the OIB ice slab detections and S-1 inferred ice slab extent in that region. We then apply those local thresholds to the entire ice sheet and calculate the F1 score by comparing the S-1 ice slab mapping to the ∼90% of the OIB observations that

were not used to choose $\alpha$, $\beta$, and $\phi$ in that trial. As with the most likely ice slab extent, we calculate separate F1 scores for the upper and lower limits of the ice slabs. From the results of these ten trials, we use the backscatter thresholds that produce the largest total ice slab area to define the maximum plausible ice slab extent, and the thresholds that produce smallest total ice slab area to define the minimum plausible ice slab extent. Figure 4 shows the results of this cross-validation. We find that across the 10 validation trials, F1 scores for the upper elevation limit of the ice slabs vary from 0.78-0.84, with no clear spatial trend. Since the F1 score for the most likely ice slab extent is 0.811, this suggest that values of $\alpha$ and $\beta$ chosen based on data from one region of the ice sheet generalize well to other regions. Indeed, these thresholds vary by only $\sim \pm 1$ dB across all regions of the ice sheet. Therefore, we assess that the algorithm is reasonably spatially robust.

We do find a clear spatial trend in the generalizability of $\phi$ between regions. In particular, when $\phi$ is derived only using data from regions NE and N1, the resulting S-1 inferred ice slab extent in Northwest and Southwest Greenland agrees poorly with the OIB observations. Conversely, the value of $\phi$ estimated using only data from the Northwest and Southwest does apply well to the North and Northeast. We suggest three explanations for this behavior. First, the North and Northeast regions have the least number of ice slab detections, so thresholds derived from data in those regions may be overfit to conditions that are not representative of larger areas. Second, snow accumulation in the North and Northeast is significantly lower than in other parts of the ice sheet, potentially leading to difference in ice slab structure and overlying snow cover. Third, we see steeper gradients in backscatter as a function of elevation in the North and Northeast compared to the Northwest and Southwest. This suggests that small variations in $\phi$ would lead to large changes in ice slab area in the Northwest and Southwest, but small changes in ice slab area in the North and Northeast. As a result, the agreement between the OIB observations and S-1 detections is much more sensitive to errors in $\phi$ in the Northwest and Southwest than in the North and Northeast.

## 4 Results and Discussion

### 4.1 Sentinel-1 Map of Ice Slab Extent

Figure 6 shows the S-1 estimated ice slab extent in winter 2016-2017, compared with the OIB ice slab detections. We find good agreement between the upper limit of the ice slabs as identified by OIB and the S-1 estimated upper limit. Figure 7 shows the confusion matrices, F1 scores, and Cohen's $\kappa$ for the minimum, most likely, and maximum S-1 estimated ice slab extent that quantify this agreement. The most likely ice slab extent has an F1 score of 0.811 with a true positive rate of 94% when detecting the upper limit of the ice slabs. However, it is important to keep in mind that the optimal values of $\alpha$, $\beta$, and $\phi$ are derived from all available ice-penetrating radar detections. Therefore, the high F1 score quantifying the agreement between the OIB detections and most likely ice slab extent mapped by S-1 simply indicates that there is a sufficiently unique relation between S-1 backscatter and firn shallow ice content that S-1 backscatter can reasonably be used as a proxy to map ice slabs. The high F1 score does not provide information on whether $\alpha$, $\beta$, and $\phi$ generalize to data collected in other places or at other times. However, the 10-fold cross validation scheme estimates $\alpha$, $\beta$, and $\phi$ using only $\sim$10% of the OIB data and validates the applicability of that threshold to the rest of the ice sheet using the withheld $\sim$90% of the data. Therefore, the minimum and

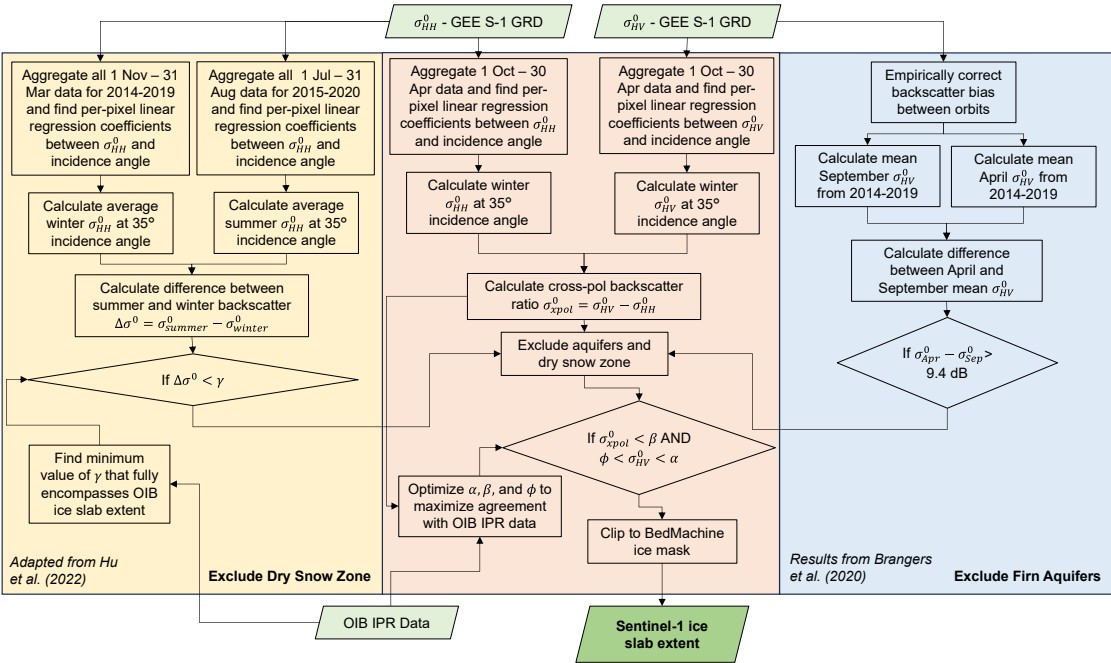

**Figure 5.** Sentinel-1 ice slab detection algorithm flowchart.

maximum ice slab extents, derived from this cross-validation scheme, show how well thresholds estimated in one region of the
ice sheet can be generalized to the ice sheet as whole.

The S-1 estimates of ice slab extent at 500 m resolution are able to capture much of the km-scale variability along the upper
elevation limit of the ice slabs, including regions of discontinuous ice slabs (see Fig. 6f for an example). The fingering structures
that we map in many regions are consistent with preferential expansion through topographic lows where water collects as it
flows laterally through saturated firn layers. Our mapping also identifies new ice slab regions in Southwest Greenland that had
not been previously classified as such in the OIB dataset, likely due to a lack of comprehensive airborne radar coverage in
this region. These newly-identified ice slab areas are highly consistent with the extent of the visible runoff zone mapped from
Landsat imagery in Tedstone and Machguth (2022), confirming that vertical percolation is limited in these areas (Fig. 8). They
are also consistent with recent firn model estimates of ice slab extent in this region (Brils et al., 2024). However, the S-1 ice
slab extent is often patchy and discontinuous in the South, likely due to the high prevalence of buried surface lakes and isolated
aquifer regions that limit detection of ice slabs due to the presence of liquid water in the subsurface.

There are also a number of discrepancies between the OIB and S-1 mapping. In the northwest, S-1 appears to slightly
underestimate the upper elevation limit of the ice slabs, particularly in the northern portion of this region, and the ice slab
extent is fairly discontinuous in this area. The S-1 algorithm generally fails to detect ice slabs in basins with persistent buried
supraglacial lakes because surface scattering from the water table dominates the return, likely contributing to this discontinuous

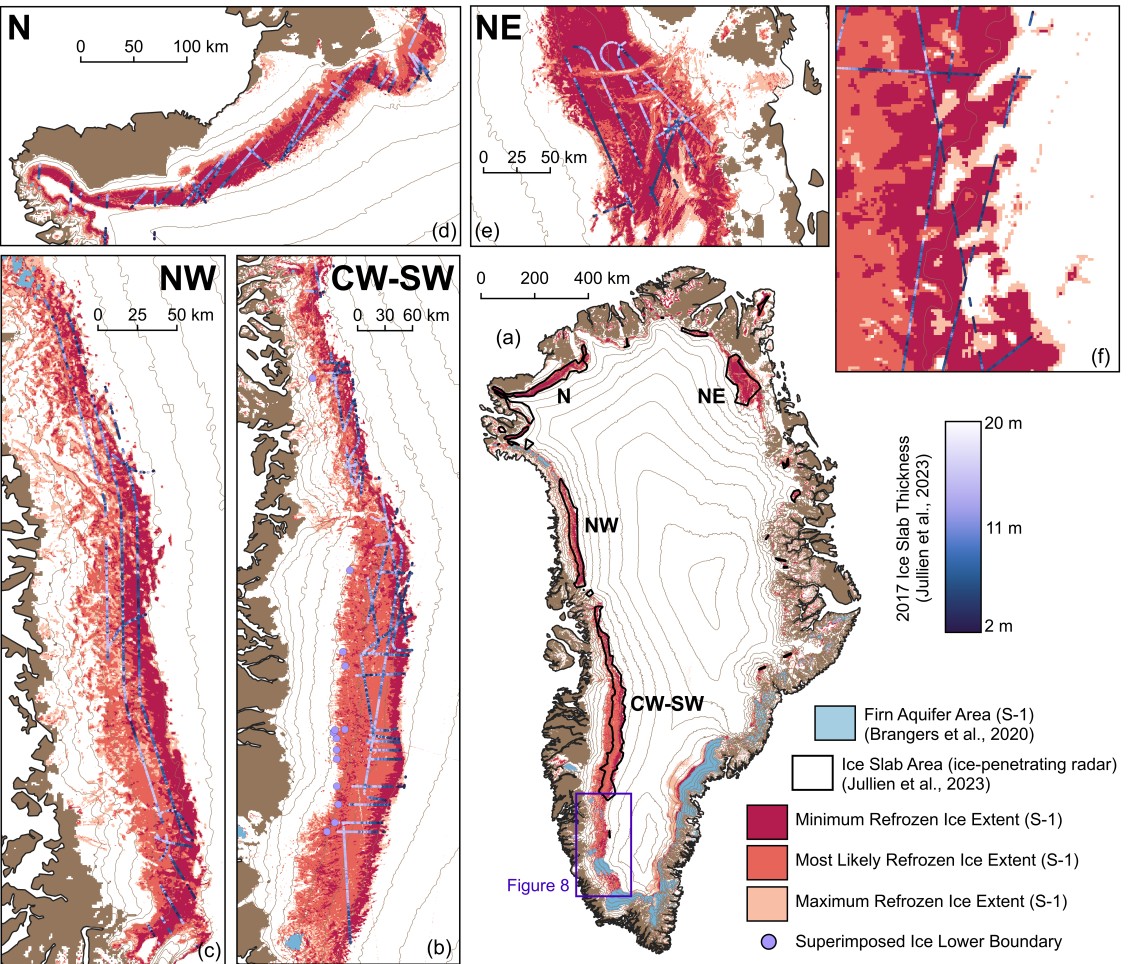

**Figure 6.** Sentinel-1 mapping of ice slabs in winter 2016-2017. a) S-1 detected ice slabs are shown in red, with the outline of the OIB detected ice slabs in the black line (Jullien, 2023). We find overall strong agreement between the S-1 and OIB mapping, although S-1 detects significant additional ice slab area in Southwest Greenland, along the Central East margin, and on peripheral ice caps. b) Zoom-in of the Central and Southwest regions. OIB ice slab detections are overlaid in the purple dots (Jullien, 2023), where darker colors indicated thinner ice slabs. There is a significant gap between the lower limit of the OIB ice slab detections and the lower limit of the S-1 mapping. The lower limit from S-1 is better aligned with the lower limit of superimposed ice as mapped from ice-penetrating radar in this paper (large purple dots). c) Zoom-in of Northwest region. d) Zoom-in of Northern region. e) Zoom-in of Northeast region. f) Zoom-in from Southwest Greenland showing details of the upper boundary. We find good agreement between the OIB and S-1 detections even where ice slabs are discontinuous due to preferential expansion in topographic lows. In all panels, the Greenland coastline was produced by the British Antarctic Survey (Gerrish, 2020), the ice mask as part of BedMachinev4 (Morlighem et al., 2017), and the 200 m contours are derived from ArcticDEM (Porter et al., 2018).

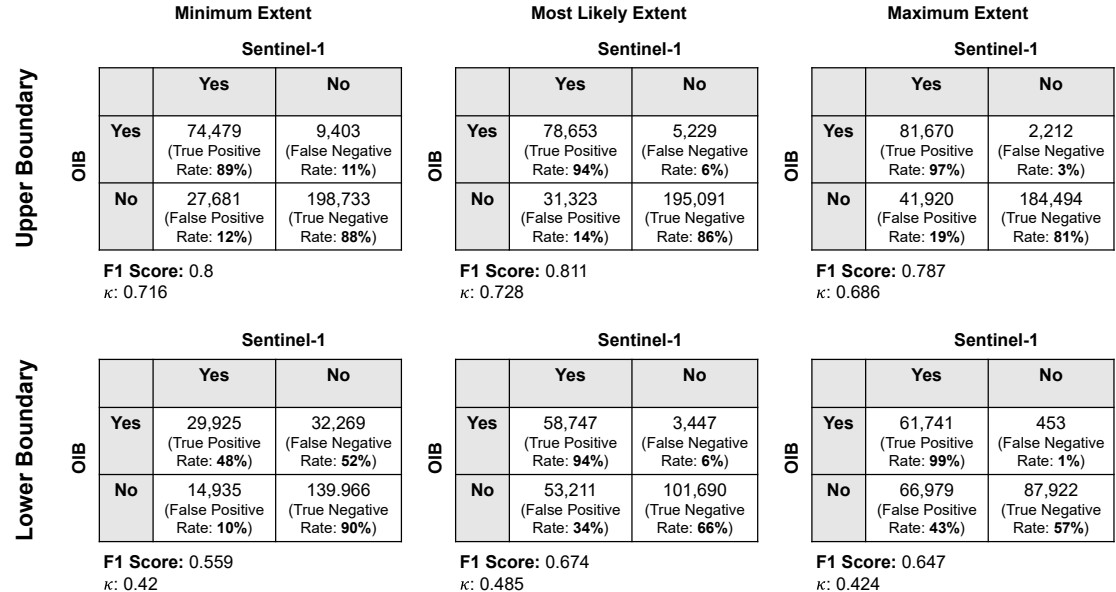

**Figure 7.** Confusion matrices quantifying the agreement between the OIB and S-1 ice slab detections for the minimum, maximum, and most likely ice slab extents. We quantify the fit for the upper boundary (top row) and lower boundary (bottom row) separately, since these thresholds were optimized separately. The most likely extent does an excellent job of detecting the upper limit of the ice slabs, with an F1 score of 0.811 and Cohen's $\kappa$ of 0.727, but the lower boundary is much more uncertain, with a best F1 score of only 0.674 and Cohen's $\kappa$ of 0.485, likely due to the consistent overestimation of ice slab extent in Southwest Greenland.

mapping in the Northwest where buried lakes are common (Koenig et al., 2015; Dunmire et al., 2021). In the Northeast, the S-1 algorithm fails to detect gaps in the ice slabs in the shear margins of the Northeast Greenland Ice Stream that are present in the OIB data. This highlights that regions with significant surface crevassing are challenging for both OIB and S-1 detection of ice slabs. S-1 will tend to overestimate ice slab extent in crevassed regions, due to enhanced $\sigma^0_{HV}$ that our algorithm ascribes to volume scattering from firn, but actually results from rough surface and multi-bounce effects within the crevasses. On the other

hand, surface crevasse clutter in the OIB data can prevent definitive classification of the near-surface structure, particularly when using radiometric metrics that assume relatively homogeneous planar structures. The S-1 algorithm also fails to detect some isolated ice slab segments identified at anomalously high elevations in the OIB data in the North and Northwest. Manual review of the radargrams in these areas shows that most fall in high melt, high accumulation areas where a thick layer of relatively transparent winter snow overlying a strong reflector at the previous summer surface may have been misclassified as

an ice slab.

    We also consistently map ice slabs along the upper boundaries of firn aquifer, both in the Northwest and Southeast, that are not identified in the OIB data. It is possible that these areas represent aquifer regions with low volumetric water content where the seasonal backscatter variability does not meet the threshold for aquifer detection, but surface scattering at the upper

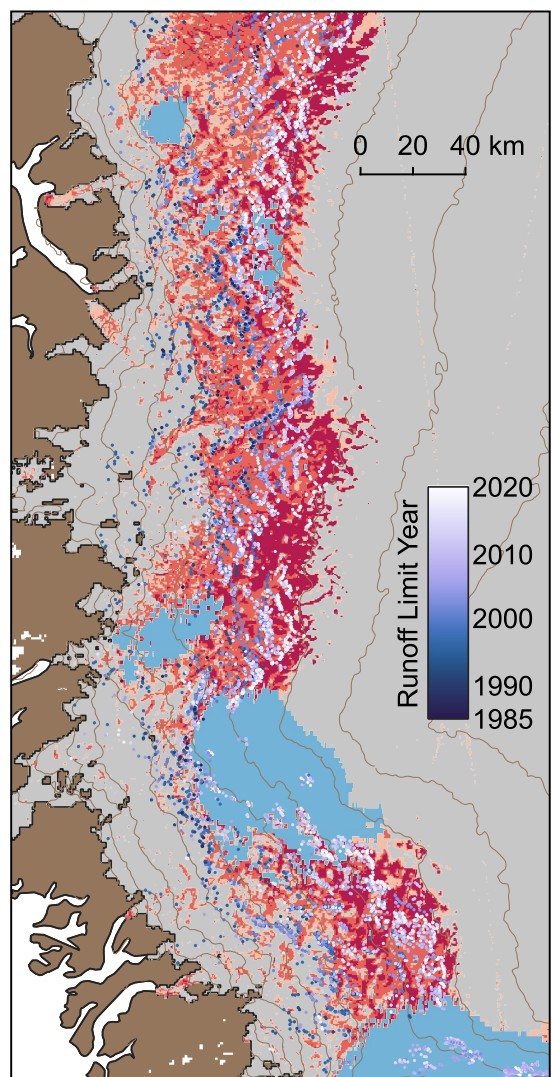

**Figure 8.** Comparison of the maximum visible runoff line from 1985-2020 with the newly mapped ice slabs regions in Southwest Greenland. The ice slab regions are marked with the same orange and red color scheme as Figure 6 and firn aquifers are shown in light blue (Brangers et al., 2020). Points marking the visible runoff limit in each sector, color-coded by year of observation, are overlaid in purple (Tedstone, 2022). There is a clear correspondence between the newly mapped ice slab regions and the runoff limit, confirming that vertical percolation is limited in these areas. The Greenland coastline was produced by the British Antarctic Survey (Gerrish, 2020), the ice mask as part of BedMachinev4 (Morlighem et al., 2017), and the 200 m contours are derived from ArcticDEM (Porter et al., 2018).

surface of the aquifer is still enhanced by partial winter meltwater retention. Time series of $\sigma^0_{HV}$ from these aquifer-marginal

areas in the Southeast show an intermediate scattering regime, with slower backscatter recovery than the percolation zone, but more rapid recovery than the well-defined aquifer regions. Alternately, there is ice-penetrating radar evidence for near-surface

refreezing in continuous ice layers less than 1 m thick following both the 2012 and 2015 melt seasons (Culberg et al., 2021; Miller et al., 2022a) that extends to the upper limit of the southeastern firn aquifers. Similar shallow ice layers might also contribute to enhanced surface scattering and lead to erroneous ice slab detections in the Southeast.

Overall, we estimate a most likely ice slab extent of 148,778 km$^2$, with a minimum ice slab extent of 59,985 km$^2$ and a maximum ice slab extent of 220,412 km$^2$. Previous estimates of total ice slab area include 60,400 - 73,500 km$^2$ from OIB data processed in Jullien et al. (2023), 76,000 km$^2$ from SMAP data processed in Miller et al. (2022a), to 230,00 km$^2$ from firn modeling (Brils et al., 2024). Much of the additional area our algorithm identifies comes from the newly detected regions in Southwest Greenland, as well as smaller contributions from narrow regions along the periphery, peripheral ice caps including

Flade Isblink, and some misclassified regions at lower elevations and in fast-flowing glacier tongues in the mountainous eastern basins. Difficulty in accurately mapping the lower boundary of the ice slabs, further discussed in Section 4.2, also adds to the discrepancy in total extent. For example, the large difference between the most likely and minimum ice slab extent is almost entirely driven by large uncertainty in the lower limit of the ice slabs. In most regions, the distance between the upper elevation limit of the minimum and most likely ice slab extents is less than ∼5 km, whereas the distance between the lower elevation

limits can range from ∼55 km in the Southwest to just ∼3 km in the North (Fig. 6).

## 4.2    Uncertainty in the Lower Boundary of Ice Slabs

Mapping the lower elevation limit of ice slabs is significantly more challenging than mapping the upper limit, as evidenced by the large uncertainty and apparently poor fit with the OIB detections. Our best estimate of the lower limit of the ice slabs has an F1 score of 0.674, compared to 0.811 for the upper boundary (Fig. 7). There are three major sources of uncertainty which

may contribute to this poor fit. First, it is likely that the limited penetration depth of S-1 prevents a clear delineation between regions where ice slabs are simply thicker than the system depth sensitivity and regions with a solid ice column. Figure 9 shows two-dimensional histograms of S-1 backscatter versus OIB-detected ice slab thickness. Both $\sigma^0_{HV}$ and $\sigma^0_{xpol}$ show little to no relationship with ice slab thickness beyond ∼7 m, suggesting that S-1 is largely insensitive to scattering structure below that depth. Since well-developed ice slabs in regions such as Southwest Greenland are often 8-10 m thick, it is unsurprising that

S-1 struggles to clearly detect the transition from ice slabs to the ablation zone. Second, the lower limit of the ice slabs in the airborne ice-penetrating radar dataset is not a data-driven boundary. Jullien et al. (2023) used the RACMOv2.3p regional climate model to exclude any regions below the long-term equilibrium line from their analysis, so the lower boundary is actually set by the model results. Given the simple snow model coupled to RACMO, the model may not accurately capture the true extent of ice slabs. Finally, there are likely regions within the upper ablation zone and lower equilibrium zone which do not

meet the formal definition of ice slabs, but where surface ice was still formed by refreezing rather than compaction. These may include regions of superimposed ice, where meltwater fully saturates the annual accumulation and refreezes to form surface ice layers (Benson, 1962), areas where the relict firn layer has been completely filled by surface meltwater draining through surface crevasses (Culberg et al., 2022a), or regions where refrozen ice was advected in from older ice slabs that formed at higher elevations. These areas would typically not be classified as ice slabs, since they lack a deep layer of relict porous firn

beneath the ice at the surface. However, since the surface ice was formed by refreezing, the C-band backscatter signatures are

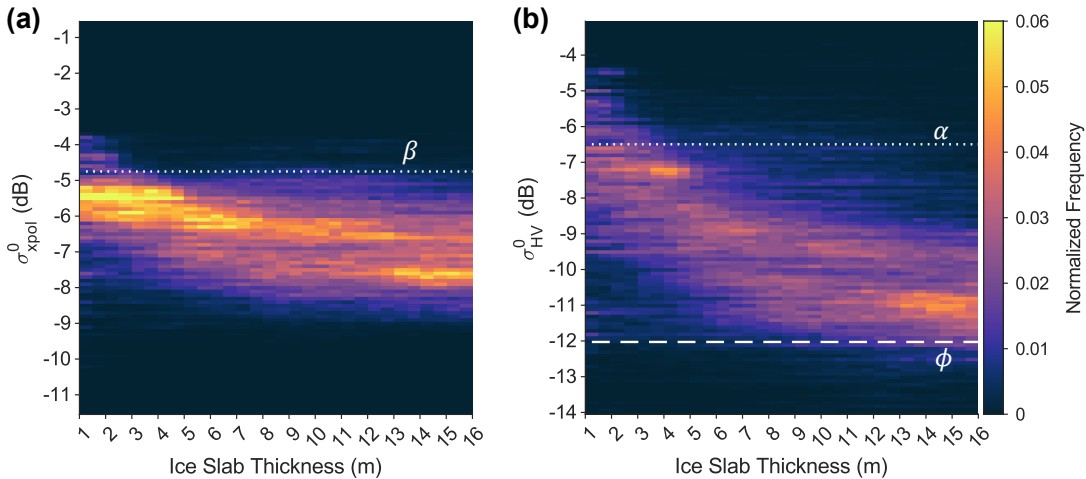

**Figure 9.** Sentinel-1 backscatter sensitivity to subsurface structure. a) Normalized two-dimensional histogram of ice slab thickness from ice-penetrating radar versus S-1 $\sigma_{xpol}^0$. b) Normalized two-dimensional histogram of ice slab thickness from ice-penetrating radar versus S-1 $\sigma_{HV}^0$. In both cases, the change in backscatter saturates around an ice slab thickness of $\sim 7$ m, suggesting that the S-1 penetration depth is limited to approximately that depth. The optimal thresholds for the upper and lower limit of the ice slabs are shown in dashed white lines on each plot. This figure also demonstrates that the $\sigma_{xpol}^0$ metric improves detection of the ice slab upper limit because the spread of backscatter values that map to an ice slab thickness of 1-2 m is significantly reduced compared to $\sigma_{HV}^0$.

likely more similar to ice slabs than to regions of the ablation zone where the surface consists of meteoric ice exhumed by ablation.

Specifically, we hypothesize that any ice formed by refreezing induces notable volume scattering due to trapped interstitial firn pockets and other heterogeneities in density, leading to a $\sigma_{HV}^0$ signature that is more similar to ice slabs than meteoric ice. This is consistent with previous work which has shown clear differences in C-band polarimetric backscatter between glacier ice, superimposed ice, and firn regions (Langley et al., 2008, 2009; Barzycka et al., 2019). To test this hypothesis, we reanalyze 14 airborne radar data flights from 2017 in Centralwest and Southwest Greenland that are approximately parallel to ice flow. Both the IMAU Firn Densification model (Brils et al., 2022) and the maximum depth of ice blobs observed in the Jakobshavn catchment (Culberg et al., 2022a) suggest that pore close-off occurs at around 30 m depth in this region. Therefore, in each radargram, we identify an continuous englacial reflector that is approximately 30 m below the surface near the upper limit of the ice slabs and assume it represents the bottom of the firn column. We trace this layer downstream until it outcrops at the surface due to ablation. Where surface sidelobes obscure the radiostratigraphy or there are significant stratigraphic disturbances near the surface, we estimate the maximum outcropping elevation as the last point where the layer can be clearly traced, and the minimum elevation as the point where we would extrapolate the layer outcropping to occur if the layer slope remained the same. Figure 10a shows an example of this layer tracing process. We infer that ice at depths shallower than the traced layer was likely formed by refreezing, rather than compaction.

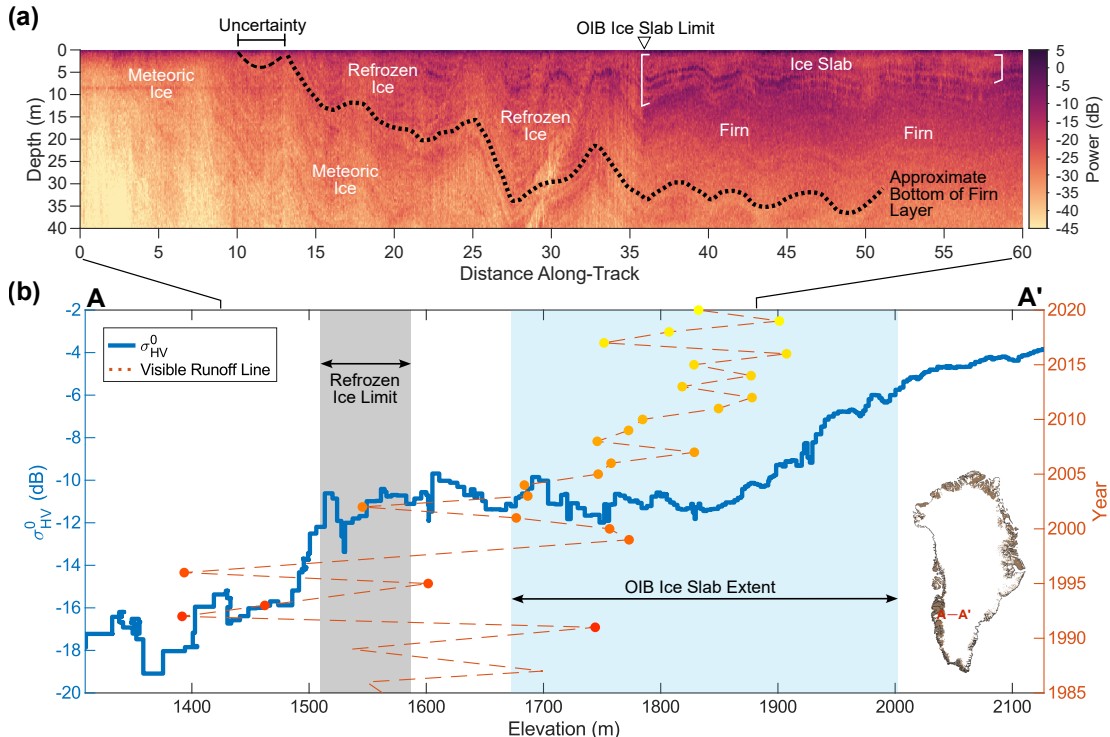

**Figure 10.** Sentinel-1 detects the lower limit of superimposed ice facies. a) Accumulation Radar transect from April 2017 (Paden et al., 2014b) showing the inferred transition from ice slabs, to superimposed ice facies, to solid meteoric ice. The dashed black line shows the englacial layer that we trace from the bottom of the firn until it outcrops at the surface in order to define the lower limit of the superimposed ice facies. b) Comparison of $\sigma^0_{HV}$ (blue line) as a function of elevation with the OIB ice slab extent (blue patch) (Jullien, 2023), estimated lower boundary of superimposed ice facies (grey patch, this paper), and the elevation of the visible runoff line between 1985 and 2020 (dashed red line with dots at annual measurement points) (Tedstone, 2022). The region where we infer that surface ice was formed by refreezing is marked by a plateau in $\sigma^0_{HV}$ around -11 dB and is also the region over which the visible runoff zone has retreated in the last two decades, supporting the idea that this region may have been near or above the firn-line in the recent past. The inset map in panel b (Gerrish, 2020; Morlighem et al., 2017) shows the location of this transect in Southwest Greenland. (Contains modified Copernicus Sentinel data 2016-2017, processed by ESA.)

In Fig. 6b, the large purple dots mark the minimum elevations of these outcropping points, showing strong agreement between the S-1 inferred lower boundary of ice slabs and this new OIB-inferred limit of superimposed ice facies. This region between the boundary of the superimposed ice facies and the lower limit of the OIB-mapped ice slabs corresponds to the area over which the visible runoff line has retreated since the mid-1980s (see Fig. 10) (Tedstone and Machguth, 2022), with significant interannual variability in runoff extent. This suggests that the S-1 mapping in part captures the historical equilibrium zone, which would have been in positive mass balance prior to the 1980s and may have still experienced intermittent years of

positive mass balance into the early 1990s. Given the slow ice flow in the Southwest ($\sim$40 ma$^{-1}$), this contributes to a wide zone where surface ice consists of old ice slabs that have not yet fully ablated, further modified by intermittent superimposed ice formation, and ongoing downstream advection of newer ice slabs. Therefore, we infer that our S-1 mapping captures not only ice slabs, but all regions where the near-surface ice was formed predominantly by refreezing.

This conclusion is consistent with some of the regional differences in the mismatch between the S-1 and OIB-inferred lower exent of the ice slabs. In the Southwest, there is a 20-35 km gap between the bottom of the OIB-detected ice slabs and S-1 mapped ice slabs. This is consistent with the low surface slopes, long history of melt, and slow and variable retreat of the snowline and expansion of the visible runoff zone in this region (Ryan et al., 2019; Tedstone and Machguth, 2022). In contrast, the two mappings agree fairly well in the North which has seen more consistent expansion of the runoff zone and retreat of the snowline since 1990 (Ryan et al., 2019; Noël et al., 2019), suggesting that the formation of extensive superimposed ice facies in this region is a more recent and rapid phenomenon. However, many of the discrepancies in the lower limit are likely attributable to other complex surface scattering mechanisms in addition to an extended superimposed ice zone. For example, in the Northwest, the S-1 lower limit is particularly diffuse, with complicated and disconnected regions identified as potential refrozen ice all the way to the ice sheet margin, particularly in the estimate of maximum ice slab extent. We hypothesize that this is due to a propensity for regions of heavy crevassing to be misclassified as refrozen ice, an issue which is more pronounced in the fast-flowing Northwest where surface strain rates are high and crevassing is prevalent.

### 4.3 Comparison with L-band Ice Slab Mapping

Figure 11 compares our S-1 derived ice slab extent with the ice slab extent derived from SMAP in Miller et al. (2022a). Overall, S-1 offers a significant improvement in both accuracy and resolution, particularly capturing regions in Northwest Greenland that SMAP failed to classify as ice slabs and accurately capturing the elevation bands where ice slabs form in the North and Northeast. However, SMAP does a somewhat better job of capturing the lower limit of the ice slabs in Southwest Greenland, in large part because the lower limit of the SMAP-inferred percolation zone (dark purple outline) is much more consistent with MODIS-inferred estimates of the summer snowline (Ryan et al., 2019) than S-1 (lilac region), which maps wet snow well into the ablation zone in some regions. SMAP also maps melt significantly further inland on the ice sheet than S-1, in part due to the comparatively coarse effective resolution, all of which contributes to different areas in which ice slabs are assumed to be viable.

The upcoming launch of the joint NASA-ISRO NISAR mission scheduled for early 2024 and eventual launch of ESA's Radar Observing System for Europe-L-Band (ROSE-L) mission will provide L-band synthetic aperture radar data with high spatial and temporal coverage over Greenland, which has the potential to offer the best of both these products. The enhanced penetration depth at L-band may particularly enable a better delineation of the low-elevation transition from ice slabs to superimposed ice. The longer wavelength will also significantly improve interferometric coherence over the ice sheet and potentially enable ice slab mapping based on volume decorrelation (Rizzoli et al., 2017) or other coherence-derived metrics. This will be a particularly important avenue of investigation given that NISAR is expected to primarily collect data in single-polarization mode over Greenland. Unfortunately, NISAR will also not collect data above 77.5° north, limiting future capacity

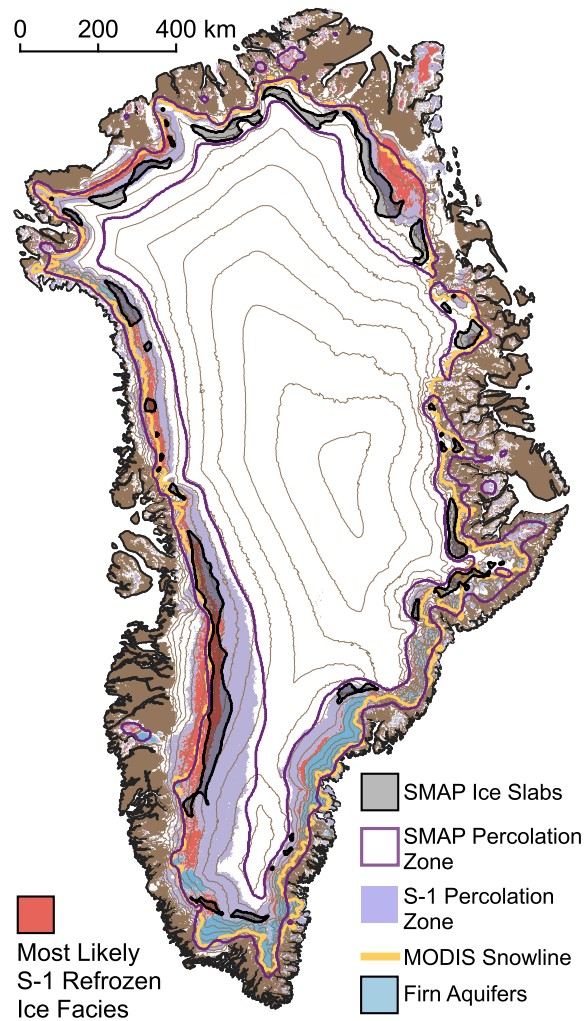

Most Likely
S-1 Refrozen
Ice Facies

SMAP Ice Slabs

SMAP Percolation
Zone

S-1 Percolation
Zone

MODIS Snowline

Firn Aquifers

**Figure 11.** Comparison of S-1 inferred ice slab extent in winter 2016-2017 (this paper) and SMAP-inferred average ice slab extent from 2015-2019 (Miller, 2021). S-1 shows a significant improvement in resolution and accuracy over SMAP. However, SMAP is able to better capture the true extent of the percolation zone, and hence the lower limit of the ice slabs, as demonstrated by the better match between the lower limit of the SMAP-derived percolation zone (Miller, 2021) and a MODIS-derived estimate of the average summer snowline (Ryan et al., 2019). Firn aquifers are shown in light blue (Brangers et al., 2020).

to study the rapidly changing northern basins. However, where data are collected, these complementary L-band observations have the potential to significantly improve our capacity to study the near-surface of Greenland from space, and our C-band algorithm development will provide an important bridge between the historical OIB data and future L-band data, which will not overlap in time with OIB. Currently, the limited data catalog of public L-band data collected by ALOS PALSAR over Greenland also offers a valuable opportunity for proof-of-concept studies that can pave the way for NISAR specific algorithms.

420

## 5 Conclusions

We have shown that Sentinel-1 winter $\sigma^0_{xpol}$ and $\sigma^0_{HV}$ signatures can be used to map the extent of Greenland's ice slabs from space at 500 m spatial resolution. Our mapping is in good agreement with both subsurface observations from the OIB ice-penetrating radar data and remote sensing observations of visible surface runoff. We identify new ice slab regions in Southwest Greenland, consistent with both firn models and runoff observations, and our mapping suggests that ice slabs are largely ubiquitous in the wet snow zone in all regions besides Southeast Greenland. Given the radiometric stability and consistent calibration efforts for Sentinel-1, we expect that it may be possible to apply the optimized thresholds we derive here for winter 2016-2017 to data collected in other years. However, there is still significant work to be done to assess the interannual radiometric stability of S-1 across the GrIS at various signal-to-noise ratios and to characterize other forms of instrumental uncertainty, particularly due to the evolving observation strategy of S-1 and missing measurements from either S-1A or S-1B in various years. Additionally, evolving conditions on the GrIS, particularly in response to extreme melt (Culberg et al., 2021) and increasing rainfall (Harper et al., 2023; Box et al., 2023), may significantly alter the subsurface stratigraphy, and therefore the observed backscatter, in ways that are not yet well-understood. Further work in required to fully characterize the physical and dielectric mechanisms that drive C-band sensitivity to firn, ice slabs, and superimposed ice structures and how their radiometric signatures may change with time. Future work might also focus on improving the discrimination of crevasses and buried or drained lakes, which can currently lead to misclassifications in ice slab regions. Regardless, the algorithm we develop here lays the groundwork for generating long time series of ice slab expansion from C-band SAR observations with sufficient spatial coverage and resolution to enable long-term monitoring and validation of predictive numerical models.

*Data availability.* Sentinel-1 mosaics (shown in Fig. 2), the final ice slab extent in winter 2016-2017 (Fig. 6), and OIB training and cross-validation data are available through Zenodo at https://doi.org/10.5281/zenodo.10892397 [Last Access: 2024-03-26]. All Sentinel-1 data were accessed and processed through Google Earth Engine. The data catalog entry can be found at https://developers.google.com/earth-engine/datasets/catalog/COPERNICUS_S1_GRD [Last Access: 2023-11-08]. Ice-penetrating radar detections of slabs are available at https://doi.org/10.5281/zenodo.7505426 [Last Access: 2023-11-08] (Jullien, 2023). Ice-penetrating radar survey lines and the radargrams shown in Figs. 1 and 10 are available from the Center for Remote Sensing and Integrated Systems at https://data.cresis.ku.edu/data/accum/ or through the National Snow and Ice Data Center at https://nsidc.org /data/iracc1b/versions/2 [Last Access: 2023-11-08] (Paden et al., 2014a, b). The elevation of the visible runoff line as a function of time is available at https://zenodo.org /records/6472348 [Last Access: 2023-11-08] (Tedstone, 2022). Sentinel-1 firn aquifer detections are available at https://arcticdata.io/catalog/view/doi%3A10.18739%2FA2HD7NS8N [Last Access: 2023-11-08] (Brangers et al., 2020). The data used throughout this paper for basemaps of Greenland are available as follows. The Greenland coastline is available from the British Antarctic Survey at https://data.bas.ac.uk/ full-record.php?id=GB/NERC/BAS/PDC/01439 [Last Access: 2023-11-08] (Gerrish, 2020). The ice mask is available through BedMachine Greenland v4 https://sites.ps.uci.edu/morlighem/dataproducts/bedmachine-greenland/ [Last Access: 2023-11-08] (Morlighem et al., 2017). The 200 m elevation contours are derived from ArcticDEM and available at https://www.pgc.umn.edu/data/arcticdem/ [Last Access: 2023-11-08] (Porter et al., 2018).

# Appendix A: Optimized Detection Thresholds

**Table A1.** Ice Slab Detection Thresholds.

| Extent | $\alpha$ | $\beta$ | $\phi$ |
|---|---|---|---|
| Minimum | -6.5 dB | -5.23 dB | -9.93 dB |
| Most Likely | -6.5 dB | -4.77 dB | -12.03 dB |
| Maximum | -5.93 dB | -3.67 dB | -13.17 dB |

## Appendix B: Spatial Resolution Sensitivity Test

Since the proposed algorithm uses backscatter thresholding to detect ice slabs, it is critical that the ice sheet mosaics primarily reflect spatial variations in backscatter due to surface properties, rather than speckle, look angle, or temporal variability. Multi-looking all of the images used in each mosaic to 500 m resolution significantly reduces speckle and improves the linear correlation between backscatter and incidence angle as a result. However, this results in some loss of spatial resolution, compared to the $\sim$ 40 m native resolution of the S-1 EW GRD product. To select the resolution used in this study, we conducted a sensitivity test on an $\sim$11,500 km$^2$ study region in Southwest Greenland that extends from the ice divide to the ice margin and includes all facies of interest, including the percolation zone, ice slab regions, and the ablation zone. For each pixel, we calculated the Pearson Correlation Coefficient between backscatter and incidence angle for both polarizations, repeating this calculation for resolutions of 50 m, 250 m, 500 m, 1 km, and 2 km. We used all images between 2016-10-01 and 2017-04-30 in these calculations. The results are shown in Fig. B1 below. We find that 500 m spatial resolution offers a good balance between high spatial resolution and reasonable speckle suppression.

## Appendix C: Linear Incidence Angle Correction

To assess the impact of the linear incidence angle correction on our results, we analyze both the residuals after correction and the impact of limiting the angular diversity in each pixel on the final ice slab classification.

### C1 Incidence Angle Correction Residuals

We calculate the incidence angle correction residuals as the root mean square error (RMSE) between the observations in each pixel and the best linear fit. The results for each polarization are shown in Fig. C1. Residuals are generally less than 1 dB for $\sigma_{HH}^0$ and 2 dB for $\sigma_{HV}^0$. On average, residuals are lowest in the interior dry snow and percolation zones and increase towards the margins, with higher and more spatially variable residuals in the ablation zone. Firn aquifers also appear as spatially contiguous regions of elevated residuals. However, we still see relatively consistent spatial variations in the slope of the correction,

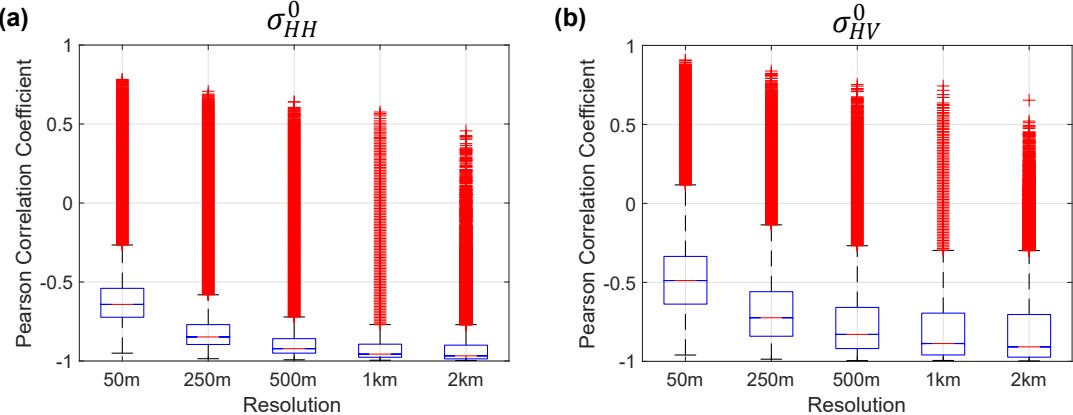

**Figure B1.** Sensitivity test for the spatial resolution of backscatter mosaics. a) Box and whisker plots showing the distribution of Pearson Correlation Coefficients between $\sigma_{HH}^0$ and incidence angle over our study area as a function of resolution. Red crosses show outliers. b) Box and whisker plots showing the distribution of Pearson Correlation Coefficients between $\sigma_{HV}^0$ and incidence angle over our study area as a function of resolution. Correlation improves as resolution degrades, with 500 m offering a reasonable balance between speckle suppression and spatial resolution.

particularly across the ablation, ice slab, and percolation zones. Slope is more variable in the interior dry snow zone, where there are also fewer observations and generally less angular diversity amongst the observations.

To better explain the source of these residuals, Fig. C2 shows scatterplots of backscatter vs. incidence angle and backscatter as a function of time for the six sites marked in white dots on Fig. C1. Sites 1-3 show behavior typical of the percolation, ice slab, and ablation zones. In Fig. C2a-f, we see that there is no strong temporal trend in backscatter over the winter season at these sites and variability is dominated by small-scale, short-term variations in backscatter that could be ascribed to accumulation, wind scour, or warming events. The linear fit between backscatter and incidence angle degrades slightly in the ablation zone, but it is not clear that a different function would provide a better correction. Overall, we assess that residuals in these regions are largely dominated by short-term variations in backscatter due to enviromental processes. In the ablation zone, surface fractures and roughness likely also play a role in modulating the relationship between backscatter and incidence angle, leading to the somewhat degraded linear fit in this region.

Sites 4-6 show regions with large residuals: Site 4 is over a firn aquifer, Site 5 is over a supraglacial lake, and Site 6 is over a heavily crevassed portion of the Jakobshavn Glacier main trunk. These sites are shown in Fig. C2g-l. We find that firn aquifer regions have large residuals because of the slow increase in backscatter with time as subsurface liquid water refreezes during the winter. This leads to a wide spread of backscatter values, so that even though there is a strong linear correlation between backscatter and incidence angle over shorter time periods, the large backscatter range leads to large residuals. Supraglacial lakes have large residuals for the same reason - there is a strong trend in backcatter with time. Crevassed regions show much more variable backscatter as a function of both time and incidence angle, likely due to the rapid ice flow and complex and anisotropic

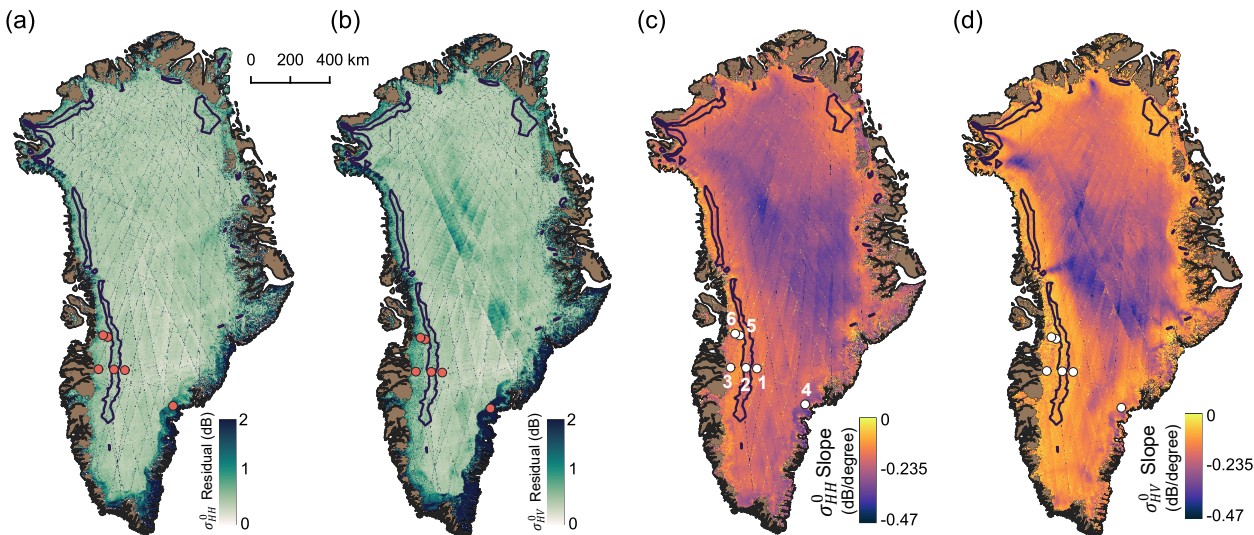

**Figure C1.** Linear incidence angle correction residuals and slopes for both polarizations. a) Root mean square error (RMSE) of the linear fit between $\sigma^0_{HH}$ and incidence angle per pixel. b) RMSE of the linear fit between $\sigma^0_{HV}$ and incidence angle per pixel. c) Slope of the linear fit between $\sigma^0_{HH}$ and incidence angle per pixel. d) Slope of the linear fit between $\sigma^0_{HV}$ and incidence angle per pixel. White and red dots mark the six sites with time series shown in Fig. C2.

roughness of the surface. However, at all these sites, the linear fit still provides a reasonable estimate of the mean winter backscatter. None of the sites show scattering behavior that could clearly be better approximated by a non-linear functional relationship between backscatter and incidence angle.

## C2 Incidence Angle Correction Sensitivity Tests

To better assess the impact of incidence angle on the final ice slab classification, we conduct a sensitivity test where we limit the range of incidence angles used to generate the backscatter mosaics. Table C1 shows the angular diversity scenarios that we explore. Limiting the range of angles used in each mosaic also has the secondary effect of reducing the total number of observations per pixel. We then optimize $\alpha$, $\beta$, and $\phi$ for each reduced mosaic using our full OIB training dataset and calculate the F1 score for each of the fifteen scenarios. In this way, we can quantify how both the angular diversity and number of observations per pixel influences the agreement between the S-1 ice slab classification and the OIB ice slab detections, as well as assess the stability of the thresholds ($\alpha$, $\beta$, and $\phi$) used for classification.

Figure C3 shows the results of this analysis. We find that the agreement between the OIB and S-1 observations converges once there are a median of $\sim$117 observations per pixel and a median angular diversity of $\sim$10° per pixel. However, if the mean incidence angle of the observations is greater than or equal to 30°, then a median of just $\sim$77 observations per pixel and an angular diversity of 7° per pixel is already sufficient. We also find that past these thresholds, any uncertainty in the final ice slab extent due to observing geometry and strategy is well within the inherent uncertainty from regional variations

**Table C1.** Range of incidence angles used for each set of test mosaics.

| Mosaic | Angles | Mosaic | Angles | Mosaic | Angles |
|--------|--------|--------|--------|--------|--------|
| 1 | 20-25 | 6 | 25-30 | 11 | 30-40 |
| 2 | 20-30 | 7 | 25-35 | 12 | 30-45 |
| 3 | 20-35 | 8 | 25-40 | 13 | 35-40 |
| 4 | 20-40 | 9 | 25-45 | 14 | 35-45 |
| 5 | 20-45 | 10 | 30-35 | 15 | 40-45 |

510    in ice slab structure and backscatter response as quantified by the range of F1 scores from the 10-fold cross-validation. Since our final ice slab classification uses the maximum number of observations and incidence angles (data points marked with the black arrows in Fig. C3), we conclude that the results are robust, since similar results are achieved with mosaics that use fewer observations or incidence angles. Figure C4 also shows that for winter 2016-2017, the area of interest over the ice slabs has at least 77 observations per pixel and an angular diversity equal to or exceeding $7°$ when all available images from 2016-10-01
515    to 2017-04-30 are used to form the backscatter mosaics.

It is notable that the F1 score shows some dependence on the median incidence angle. For example, mosaics formed with incidence angles from $20°-25°$ perform significantly worse that mosaics formed with incidence angles from $35°-40°$, despite having similar angular diversity and number of observations per pixel. This suggests that backscatter measurements at higher incidence angles provide better discrimination of ice slabs, consistent with the theoretical prediction that the largest difference
520    in backscatter between a smooth surface and a volume scattering medium should occur at large incidence angles. Therefore, it is important the the majority of observations be collected at incidence angles greater than $30°$ for accurate ice slab classification. This criterion is met when using all available images between 2016-10-01 and 2017-04-30.

We also considered the convergence of the detection thresholds $\alpha$, $\beta$, and $\phi$ themselves. These results are shown in Fig. C5. We find that the thresholds converge quickly and fall comfortably within the range of plausible thresholds inferred from the
525    10-fold cross-validation. We conclude that these thresholds are robust when all available data is used, and that the uncertainty introduced by spatial variations in the number of observations or range of incidence angles is less than the uncertainty from spatial variations in ice slab structure.

*Author contributions.*    RC conceived the study, designed and implemented the processing algorithms, and wrote the initial paper draft. RJM and JZM provided guidance and suggestions on the Sentinel-1 processing and ice slab detection workflow. All authors contributed to the
530    scientific analysis of results and writing of the final manuscript.

*Competing interests.*    The contact author has declared that none of the authors has any competing interests.

*Acknowledgements.* JZM was supported by NASA grants 80NSSC20K1806 and 80NSSC21K0749.

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

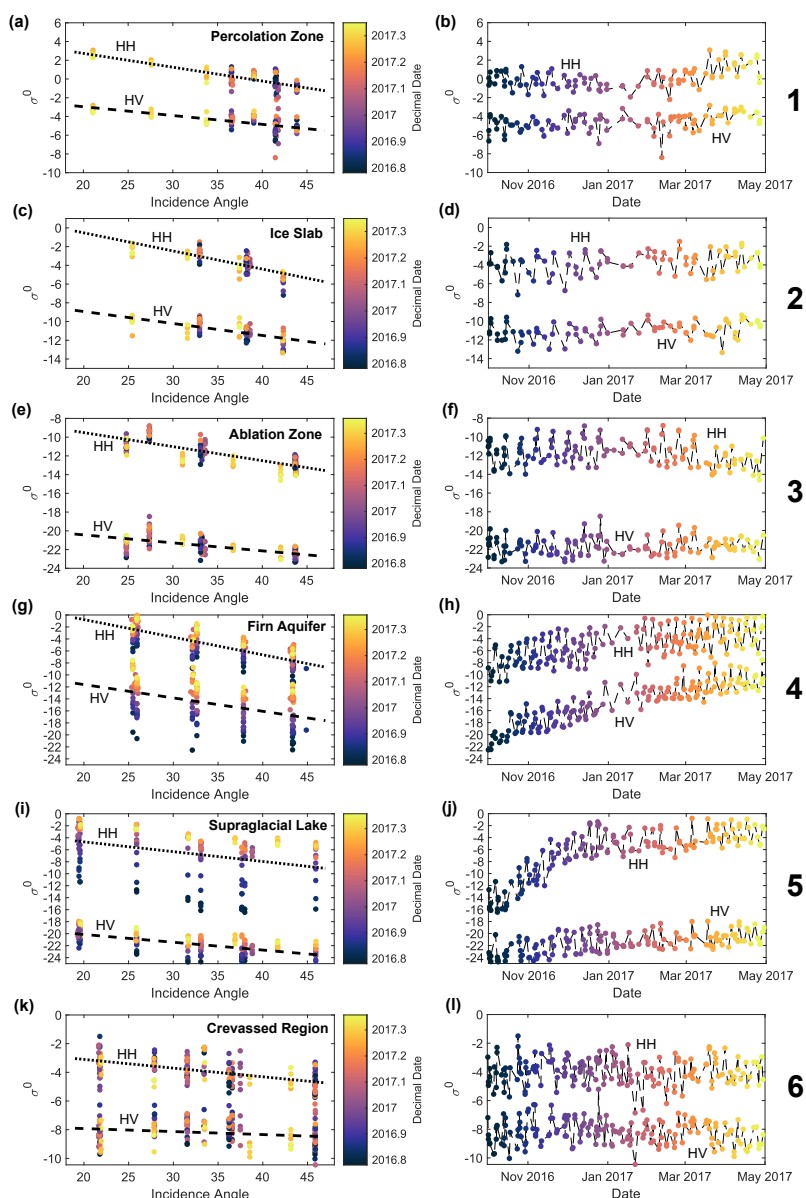

**Figure C2.** a) Scatterplot of backscatter vs. incidence angle at Site 1 in the percolation zone for both polarizations. b) Scatterplot of backscatter vs. time from 2016-10-01 to 2017-04-30 at Site 1. c) Scatterplot of backscatter vs. incidence angle at Site 2 in the ice slab zone for both polarizations. d) Scatterplot of backscatter vs. time from 2016-10-01 to 2017-04-30 at Site 2. e) Scatterplot of backscatter vs. incidence angle at Site 3 in the ablation zone for both polarizations. f) Scatterplot of backscatter vs. time from 2016-10-01 to 2017-04-30 at Site 3. g) Scatterplot of backscatter vs. incidence angle at Site 4 in a firn aquifer area for both polarizations. h) Scatterplot of backscatter vs. time from 2016-10-01 to 2017-04-30 at Site 4. i) Scatterplot of backscatter vs. incidence angle at Site 5 over a supraglacial lake for both polarizations. j) Scatterplot of backscatter vs. time from 2016-10-01 to 2017-04-30 at Site 5. k) Scatterplot of backscatter vs. incidence angle at Site 6 over a heavily crevassed region for both polarizations. l) Scatterplot of backscatter vs. time from 2016-10-01 to 2017-04-30 at Site 6.

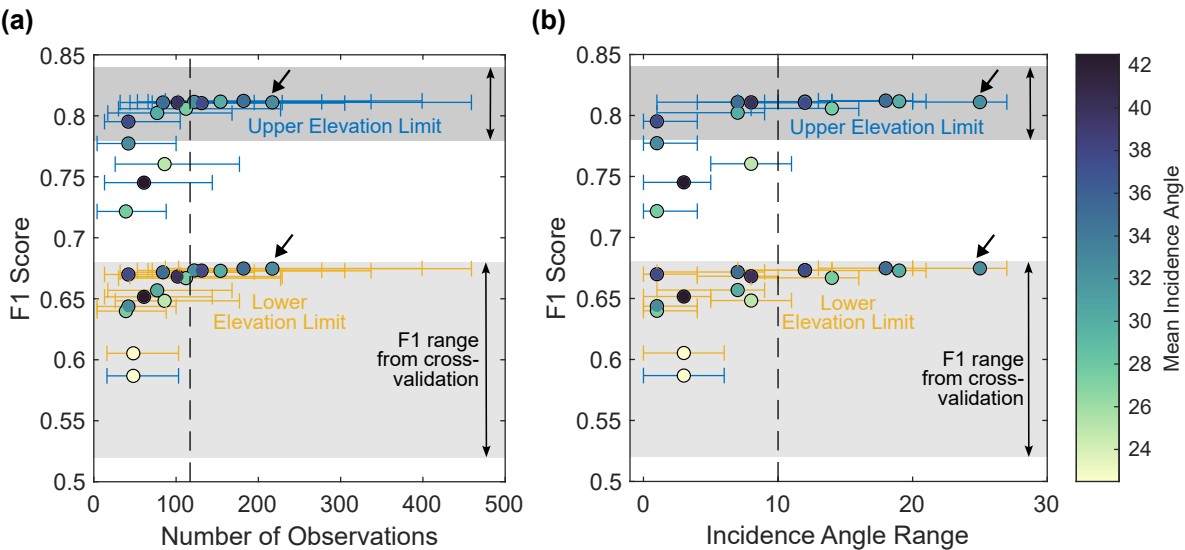

**Figure C3.** Convergence of final ice slab classification results. a) F1 scores for the upper elevation limit of the ice slabs (blue) and lower elevation limit (yellow) as a function of the median number of observations per pixel. Error bars show the 5th and 95th percentile of the observation count over the whole ice sheet. The color of each dot shows the median incidence angle used for that mosaic. Grey patches show the range of F1 scores from the 10-fold cross-validation, quantifying the inherent spatial variability in agreement between the S-1 and OIB ice slab classifications. Black arrows mark the mosaic used for the final ice slab classification in the main paper. The dashed line shows the approximate breakpoint where the F1 scores converge. b) F1 scores for the upper elevation limit of the ice slabs (blue) and lower elevation limit (yellow) as a function of the true angular diversity of observations (number of unique incidence angles) per pixel. Error bars shown the 5th and 95th percentile of the angular diversity over the whole ice sheet. All other plot components are as described in the panel (a) caption.

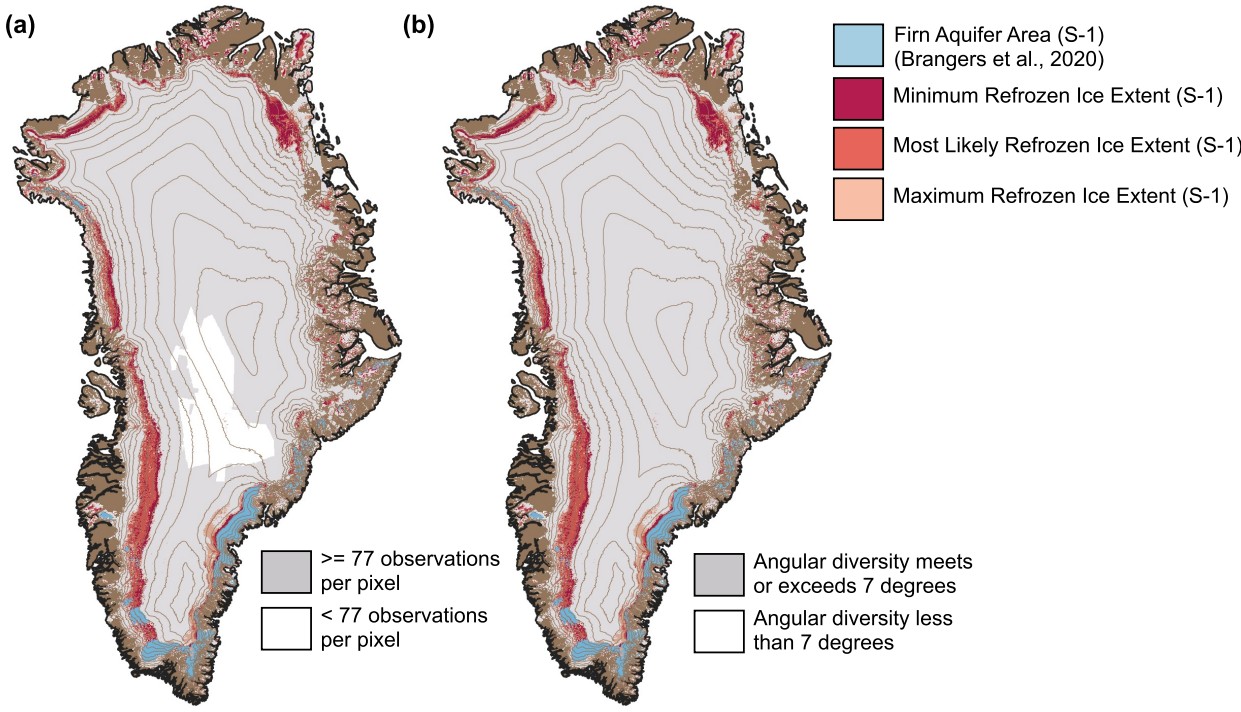

**Figure C4.** The fraction of the Greenland Ice Sheet that meets the convergence thresholds for number of observations per pixel and angular diversity per pixel when considering all data collected between 2016-10-01 and 2017-04-30. a) Number of observations per pixel. Regions shaded in light grey meet or exceed the threshold of 77 observations per pixel. All ice slab regions meet this threshold, with only a small portion of the interior lacking sufficient observations. b) Angular diversity per pixel. Regions shaded in light grey have an angular diversity of at least 7°. The criterion is met for the entire ice sheet. S-1 detected ice slabs are shown in shades of red and orange to emphasize that these regions have robust data coverage.

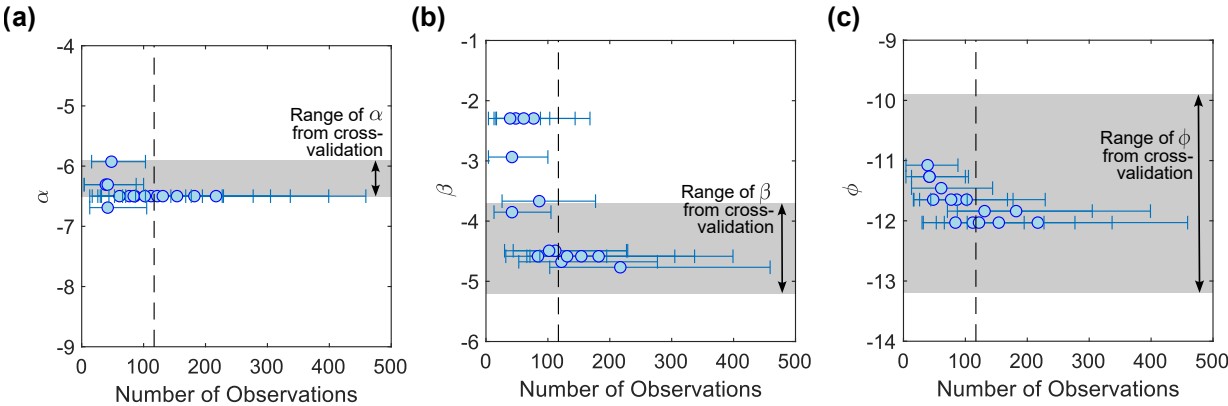

**Figure C5.** Convergence of classification thresholds as a function of the number of observations (the results are analogous for angular diversity). a) $\alpha$ as a function of the median number of observations per pixel. Error bars show the 5th and 95th percentile of the observation count over the whole ice sheet. Grey patches show the range of F1 scores from the 10-fold cross-validation, quantifying the inherent spatial variability in the optimal threshold. The dashed line shows the approximate breakpoint where the threshold converges. b) $\beta$ as a function of the median number of observations per pixel. c) $\phi$ as a function of the median number of observations per pixel.