# Peer review of "Sentinel-1 Detection of Ice Slabs on the Greenland Ice Sheet"

_EGUsphere, 2023_

## Referee Comment (RC1)

Review of "Sentinel-1 Detection of Ice Slabs on the Greenland Ice Sheet"
Culberg et al.

This manuscript describes a new algorithm which uses Sentinel-1 backscatter observations to detect ice slabs across the Greenland Ice Sheet. The work is new and interesting and will help fill important scientific gaps by providing a method to detect ice slabs at higher spatial and temporal resolution. Overall the paper is clear and well written and has high-quality figures. Once my concerns are addressed, I believe this paper will be an excellent contribution.

**Major comments**
1) I understand that the authors take into account different incidence angles for the Sentinel-1 data by applying a linear fit to incidence angle and backscatter; however, I feel like the impact of incidence angle on this work needs to be more fully understood before this algorithm can be applied. How many different incidence angles are available for each pixel? If one pixel has substantially more incidence angles available, how does this impact the cross-pol ratio and therefore the delineation of ice slabs? It would be interesting to investigate how the defined ice slab boundaries change if only some of the available incidence angles in a given region were used. This would shed some insight into how sensitive this algorithm is to various incidence angles.
2) I am also a bit concerned (or maybe just confused) about the testing of the algorithm. In the abstract, the authors state "The S-1 inferred ice slab extent is in excellent agreement with ice penetrating radar ice slab detections from spring 2017". However, I find this to be misleading since the training dataset was from spring 2017. Of course the S1 ice slab extent is in good agreement since the algorithm seems to be empirically derived from this data. Was there a completely independent dataset used to test the algorithm? Can it be tested with OIB data from a different year? It seems that the F1 scores given in lines 205-208 and Figure 7 were from the training dataset.
3) I am also a bit confused with how the folds were created. Were these folds selected completely randomly or separated by specific regions of the ice sheet. If the latter, this could provide insight into the spatial robustness of this algorithm. For example, in Figure 4, which region corresponds with fold 2 and why is the F1 score for the lower elevation limit so much worse in this region? The authors state in the caption of Figure 4 that "we discard the iteration marked with the red bar due to anomalously poor F1 score…" but it seems like this anomalously low F1 score should be important as it says something about the robustness of the algorithm? This should be further explored and discussed.

**Minor comments**
1) In paragraph 1 of the introduction please also mention that mass is also lost due to dynamical processes and it would be helpful to briefly compare this mass loss to that from surface processes.
2) L28: "preferences the formation of perennial firn aquifers" is a bit awkward wording.
3) L42: "… including the first high elevation rain event, such as 2019, 2021, and 2023." This wording makes it sound like the rain event occurred in 2019, 2021, and 2023.

4) L111: From Fig. 1, it looks like the HV backscatter is closer to -4 dB in the percolation zone.
5) L112: "…eventually reaching a plateau around -11 dB". This is a bit misleading I think. There is still substantial variation around this new plateau as the HV backscatter changes from -8 dB in the upper part of the ablation zone to -13 dB in the wet snow zone.
6) L191: "… we optimize independent backscatter thresholds…" What are the thresholds independent from?
7) L196: What is meant by "high-end estimate"?
8) L202: what step size did you use to test $\alpha$ and $\beta$ within these ranges?
9) After the 10-fold validation, how were the optimal empirical parameters chosen?
10) L243: Please add Dunmire et al 2021 with Koenig et al 2015 citation.
11) Lines 285-290: I find this section confusing. Isn't "ice formed be refreezing" (L286) the same as an "ice slab"? The distinction between ice slabs and other refrozen ice is unclear throughout this section. Also, it seems that "ice formed by refreezing induces significant volume scattering due to trapped air bubbles…" (L286) contradicts the introduction "with relatively little volume scattering since heterogeneities such as air bubbles are significantly smaller than the C-band wavelength" (L95).

**Technical corrections**
1) L22: Please add "meltwater" before "retention and runoff"
2) L32: Please add "elevation" in "upper *elevation* limit" (also for L194).
3) L150: Delete "to" before "correction" at the start of this line.

---

## Referee Comment (RC2)

Review of "Sentinel-1 Detection of Ice Slabs on the Greenland Ice Sheet"

This paper presents an investigation of the potential for detecting ice slabs in the Greenland Ice Sheet using Sentinel-1 HH and HV C-band radar backscatter data. The paper is interesting and seems to show some promise for the method. The authors provide an appropriate degree of assessment indicating the regions of most uncertainty. I believe the paper can be published after some revisions, as follows:

1) Why was 500 m resolution used? The authors never discuss this. Finer resolution would be of interest. Was there a reason it was not pursued?

2) The authors discuss their reasoning for using only 1 year of data in the paper's conclusions. I can see their points, but the paper would be more impactful if a multi-year study were performed. I recommend at a minimum that the authors describe their reason for using only a single year earlier in the paper.

3) Bottom of p. 2: should say radiometer not radar data. 2$^{nd}$ paragraph of p. 3: dielectric misspelled

4) Could the authors provide more information on the residual errors after the angle correction is performed, e.g. a plot or two of the data before and after angle correction? I'm assuming there is a lot of residual here due to the highly variable topography, etc. which makes such effects also a potential source of error that could be included in later discussions.

5) Figure 2 could be more appealing if zoomed somehow in the manner of Figure 6. As is, we mostly see the interior ice sheet regions that are not of interest in this study. Perhaps rotate 90 degrees and separate into rotated images of the East and West coasts of the ice sheet?

6) Was the same angle correction used for the Summer data? If so, this should also be noted as a potential (minor) issue.

7) The apparent alignment of the contours in Figure 3 with a "45 degree line" suggests that a classifier based on sigma_HH + alpha sigma_HV where alpha is some constant might also be successful here, rather that the separate thresholds on each quantity?

8) Figure 5 appears somewhat redundant given that everything has been described in the text by this point.

9) I found the distinction between the "minimum", "maximum", etc. classifiers somewhat confusing. Please introduce these ideas more clearly early on in the discussion of the classifier.

10) Dashed lines in Figures 6 and 11 are hard to discern especially when the topographic lines are included. Is there a way of doing this that makes them more clear? This comparison is key so making it easy to follow is crucial.

11) Some of the discussions of differences seemed a little long and overly complex. Consider trying to simplify these discussions if possible, i.e. simply state "may have more volume scattering compared to xxx" etc.

---

## Author Response (AR1)

**Response to Editor Comments**

*Dear Riley Culberg,*

*Thank you for your comments to the reviewers. The three reviewers think your paper is new and interesting. It provides a new method to detect Greenland ice slab from Sentinel-1 imagery and may "fill important scientific gaps" in Greenland hydrology. They also provide some important comments to further improve the quality of your paper. Particularly, the impact of different incidence angles, the validation of the algorithm, and the spatial resolution of the data product should be further demonstrated. I agree with these comments and encourage you to take into account these comments in the revised paper.*

*Best,*

*Dr. Kang Yang*
*Editor, The Cryosphere*

Thank you for handling our manuscript! In response to the reviewer comments you highlighted, we have:

[1] Conducted an extensive sensitivity analysis on the impacts of angular diversity and number of observations per pixel on the final ice slab classification results and threshold choices. Details are provided in the new Supplementary Information and demonstrate that the thresholds and F1 score converge for a smaller degree of angular diversity and lower number of observations than is used in our final product. Therefore, the finally mapping is as robust as possible given the available observations.

[2] Thoroughly rewritten Section 3.3 detailing our approach to the validation of our results and characterizing their uncertainty to better highlight why we chose the 10-fold cross-validation method and how it works. We have also added caveats to our description of the results to highlight how they should be interpreted in light of our validation technique.

[3] Conducted a sensitivity analysis on the spatial resolution which is detailed in the new Supplementary Information. This analysis demonstrates that our chosen resolution of 500 m provides the appropriate balance between high spatial resolution and the necessary speckle suppression.

Below we provided a point-by-point response to all the reviewers' comments, including pointers to where we have made changes in the manuscript. We believe these comments have significantly improved our work.

**Response to Reviewer #1**

*This manuscript describes a new algorithm which uses Sentinel-1 backscatter observations to detect ice slabs across the Greenland Ice Sheet. The work is new and interesting and will help fill important scientific gaps by providing a method to detect ice slabs at higher spatial and temporal resolution. Overall, the paper is clear and well written and has high-quality figures. Once my concerns are addressed, I believe this paper will be an excellent contribution.*

Thank you! We appreciate your thoughtful and constructive comments that will help us improve this manuscript.

**Major comments**

**[R1]** *I understand that the authors take into account different incidence angles for the Sentinel-1 data by applying a linear fit to incidence angle and backscatter; however, I feel like the impact of incidence angle on this work needs to be more fully understood before this algorithm can be applied. How many different incidence angles are available for each pixel? If one pixel has substantially more incidence angles available, how does this impact the cross-pol ratio and therefore the delineation of ice slabs? It would be interesting to investigate how the defined ice slab boundaries change if only some of the available incidence angles in a given region were used. This would shed some insight into how sensitive this algorithm is to various incidence angles.*

Thanks for this interesting suggestion! We have conducted this sensitivity analysis and describe the results below. In our revised manuscript, we will add these sensitivity analysis results and conclusions to a new Supplementary Information file.

In EW mode, Sentinel-1 incidence angles from 18.9-47 degrees. We generated 15 $\sigma_{HV}$ and $\sigma_{HH}$ backscatter mosaics of the Greenland Ice Sheet using subsets of these incidence angles. The full list of mosaics is given in the table below. This has the two-fold effect of limiting the angular diversity per pixel, as well as the total number of observations per pixel.

**TABLE I**

| Mosaic # | Angles | Mosaic # | Angles | Mosaic # | Angles |
|---|---|---|---|---|---|
| 1 | 18.9-25 | 6 | 25-30 | 11 | 30-40 |
| 2 | 18.9-30 | 7 | 25-35 | 12 | 30-37 |
| 3 | 18.9-35 | 8 | 25-40 | 13 | 35-40 |
| 4 | 18.9-40 | 9 | 25-47 | 14 | 35-47 |
| 5 | 18.9-47 | 10 | 30-35 | 15 | 40-47 |

For each of these fifteen mosaics, we optimized the ice slab detection thresholds ($\alpha, \beta,$ and $\phi$) using the full 2017 OIB dataset and calculated the F1 score quantifying the agreement between the OIB observations and the S-1 ice slab detections. We only considered pixels with observations from more than one incidence angle. The figures below show how the agreement between the OIB observations and S-1 detections changes as a function of the number of observations per pixel and the angular diversity of those observations over our region of interest. The median number of observations or angular range (difference between true minimum and maximum incidence angles in each pixel) for each mosaic are show in dots. The dots are colored by the mean incidence angle in the range. (For example, mosaics 6 and 10 both have an incidence angle range of no more than 5°, but mean incidence angles of 27.5° and 32.5° respectively). Error bars indicate the 5th and 95th percentiles of those values. Blue bars show how the F1 score for the upper elevation limit of the ice slabs changes, while yellow bars show the F1 score for the lower limit of the ice slabs. Grey patches

show the range of F1 values reported from the 10-fold cross-validation scheme, and therefore quantify the variability in agreement between OIB and S-1 detections that we observe when optimizing $\alpha$, $\beta$, and $\phi$ using subsets of the OIB data from different regions of the ice sheet.

[Figure]

**Figure 4.** Convergence of final ice slab classification results. a) F1 scores for the upper elevation limit of the ice slabs (blue) and lower elevation limit (yellow) as a function of the median number of observations per pixel. Error bars shown the 5th and 95th percentile of the observation count over the whole ice sheet. The color of each dot shows the median incidence angle used for that mosaic. Grey patches show the range of F1 scores from the 10-fold cross-validation, quantifying the inherent spatial variability in agreement between the S-1 and OIB ice slab classifications. Black arrows mark the mosaic used for the final ice slab classification in the main paper. The dashed line shows the approximate breakpoint where the F1 scores converge. b) F1 scores for the upper elevation limit of the ice slabs (blue) and lower elevation limit (yellow) as a function of the true angular diversity of observations (number of unique incidence angles) per pixel. Error bars shown the 5th and 95th percentile of the angular diversity over the whole ice sheet. All other plot components are as described in the panel (a) caption.

From these results, we see that the agreement between the OIB and S-1 observations converges once we have a median of ~117 observations per pixel and a median angular diversity of ~10 degrees per pixel. We also find that past these thresholds, any uncertainty in the final ice slab extent due to variations in the number of observations or angular diversity of observations in each pixel is well within the inherent uncertainty from regional variations in ice slab structure and backscatter response as quantified by the range of F1 scores from the 10-fold cross-validation. Since the original manuscript uses the maximum number of observations and incidence angles (equivalent to mosaic 5, or the furthest right data point on both plots), we conclude that the results are robust, since similar results are achieved with mosaics that use fewer observations or incidence angles.

We also considered the convergence of the detection thresholds $\alpha$, $\beta$, and $\phi$ themselves. Plots for these results are shown in the figures below as a function of number of observations (the plots for angular diversity are essentially the same). Similarly, we see that the thresholds converge fairly quickly and fall comfortably within the range of plausible thresholds inferred from the 10-fold cross-validation. We conclude that these thresholds are robust when all available data is used, and that the uncertainty introduced by spatial variations in the number of observations or range of incidence angles is less than the uncertainty from spatial variations in ice slab structure. We think that the minimum and maximum ice slab extent shown in the original manuscript adequately quantifies the uncertainty in ice slab extent, since it is derived from the minimum and maximum thresholds that define the grey boxes in the plots, which fully encompass the variability from

observational geometry.

[Figure]

**Figure 5.** Convergence of classification thresholds. These are shown as a function of the number of observations, but the results are analogous for angular diversity. a) $\alpha$ as a function of the median number of observations per pixel. Grey patches show the range of F1 scores from the 10-fold cross-validation, quantifying the inherent spatial variability in the optimal threshold. The dashed line shows the approximate breakpoint where the threshold converges. b) $\beta$ as a function of the median number of observations per pixel. c) $\phi$ as a function of the median number of observations per pixel.

**[R2]** *I am also a bit concerned (or maybe just confused) about the testing of the algorithm. In the abstract, the authors state "The S-1 inferred ice slab extent is in excellent agreement with ice penetreting radar ice slab detections from spring 2017". However, I find this to be misleading since the training dataset was from spring 2017. Of course the S1 ice slab extent is in good agreement since the algorithm seems to be empirically derived from this data. Was there a completely independent dataset used to test the algorithm? Can it be tested with OIB data from a different year? It seems that the F1 scores given in lines 205-208 and Figure 7 were from the training dataset.*

You are correct that we do not use independent training and validation sets to estimate the optimal backscatter thresholds for mapping ice slab extent. When we say that the S-1 mapping and OIB detections are excellent agreement, we simply mean to highlight that we have shown that the S-1 backscatter has a strong relationship with near-surface ice content and therefore can be used to map the presence of ice slabs. Prior to this study, we did not know that this would necessarily be the case. Certainly, we can see from the issues with detecting the lower elevation limit of the ice slabs that even when using all available data to choose our empirical thresholds, there is still the potential for significant discrepancies between the airborne ice-penetrating radar detections and the best-fit S-1 mapping.

We considered using the smaller but independent dataset of airborne ice-penetrating radar detections from spring 2018 for validation. However, we felt that this would add significant complications to the manuscript, since we would need to address the interannual radiometric stability of S-1 and environmental conditions other than ice slabs that might drive interannual variability in mean winter backscatter to fully interpret the validation results. While we think this can be done, and it the obvious next step, we think it is beyond the scope of this paper which is focused on investigating whether S-1 is capable of or appropriate for mapping ice slabs at the ice-sheet scale in the first place.

Since we do not introduce independent data from a different year, we use the 10-fold validation scheme used to assess uncertainty in the optimal thresholds (and the resulting most likely ice slab extent), which is why we also provide the confusion matrices and F1 score for the minimum and

maximum ice slab extent. In the 10-fold cross-validation, we only use ~10% of the data to select the thresholds and then evaluate the results on their agreement with the remaining ~90% of the data that was not used for estimating the thresholds. So, in this process, we have independent training and validation sets.

We have made some key changes throughout the manuscript to clarify our approach and avoid overstating our results in the final paper. Key changes are detailed below:

[1] The abstract reads as below, where amended text is noted in bold.
*"Ice slabs are multi-meter thick layers of refrozen ice that limit meltwater storage in firn, leading to enhanced surface runoff and ice sheet mass loss. To date, ice slabs have largely been mapped using airborne ice-penetrating radar, which has limited spatial and temporal coverage. This makes it difficult to fully assess the current extent and continuity of ice slabs or to validate predictive models of ice slab evolution that are key to understanding their impact on Greenland's surface mass balance. Here, for the first time, we map the extent of ice slabs and similar superimposed ice facies across the entire Greenland Ice Sheet at 500 m resolution using dual-polarization Sentinel-1 (S-1) synthetic aperture radar data collected in winter 2016-2017*. **We do this by selecting empirical thresholds of the cross-polarized backscatter ratio and HV backscattered power that optimize agreement between airborne ice-penetrating radar data detections of ice slabs and the S-1 estimates of ice slab extent. Our results show that there is a sufficiently strong relation between C-band backscatter and the ice content of the upper ~7 meters of the firn column to enable ice slab mapping with S-1. We find that ice slabs are nearly continuous around the entire margin of the ice sheet. This includes regions in Southwest Greenland where ice slabs have not been previously identified, but where the S-1 inferred ice slab extent is in excellent agreement with the extent of visible runoff mapped from optical imagery.** *The algorithm developed here lays the groundwork for long-term monitoring of ice slab expansion with current and future C-band satellite systems and highlights the added value of future L-band missions for near-surface studies in Greenland."*

[2] In Section 4.1, when introducing our results, we explicitly discuss how the F1 score can be interpreted. Bolded text below was added in this paragraph (lines 279-287 in the revised manuscript)/
*"Figure 6 shows the S-1 estimated ice slab extent in winter 2016-2017, compared with the OIB ice slab detections. We find* **strong** *agreement between the upper limit of the ice slabs as identified by OIB and the S-1 estimated upper limit. Figure 7 shows the confusion matrices, F1 scores, and Cohen's $\kappa$ for the minimum, most likely, and maximum S-1 estimated ice slab extent that quantify this agreement. The most likely ice slab extent has an F1 score of 0.811 with a true positive rate of 94% when detecting the upper limit of the ice slabs.* **However, it is important to keep in mind that the optimal values of $\alpha$, $\beta$, and $\phi$ are derived from all available ice-penetrating radar detections. Therefore, the high F1 score quantifying the agreement between the OIB detections and most likely ice slab extent mapped by S-1 simply indicates that there is a sufficiently unique relation between S-1 backscatter and firn shallow ice content that S-1 backscatter can reasonably be used as a proxy to map ice slabs. The high F1 score does not provide information on whether $\alpha$, $\beta$, and $\gamma$ generalize to data collected in other places or at other times. However, the 10-fold cross validation scheme estimates $\alpha$, $\beta$, and $\gamma$ using only ~10% of the OIB data and validates the applicability of that threshold to the rest of the ice sheet using the withheld ~ 90% of the data. Therefore, the minimum and maximum ice slab extents, derived from this cross-validation scheme, show how well thresholds estimated in one region of the ice sheet can be generalized to the ice sheet as whole."*

[3] In the revised manuscript, we have added Section 3.3.2 (lines 249-272) that discusses regional

variations that could lead to the variations in $\alpha$, $\beta$, and $\phi$ that we see when the thresholds are selected based on data from a particular region (see response to R3 below).

[4] We have clarified our explanation of the 10-fold cross-validation scheme to highlight why it is implemented and how we use it in place of an independent validation data set to quantify uncertainty in our results. The full text of these revisions encompasses Section 3.3 in the revised text.

**[R3]** *I am also a bit confused with how the folds were created. Were these folds selected completely randomly or separated by specific regions of the ice sheet. If the latter, this could provide insight into the spatial robustness of this algorithm. For example, in Figure 4, which region corresponds with fold 2 and why is the F1 score for the lower elevation limit so much worse in this region? The authors state in the caption of Figure 4 that "we discard the iteration marked with the red bar due to anomalously poor F1 score…" but it seems like this anomalously low F1 score should be important as it says something about the robustness of the algorithm? This should be further explored and discussed.*

Thank you – this is a very constructive suggestion on designing the folds! We now split the training data set into 10 distinct regions of the ice sheet. This leads to small variations in the size of each validation fold (e.g. total number slab and no slab observations in each fold), but allows us, as you suggested, to interrogate the spatial robustness. Results are shown in the figure below. We have updated the final manuscript to include this new division of the folds and included a section discussing the regional differences we observe (Section 3.3.2).

We find that the algorithm is quite spatial robust when delineating the upper elevation limit of the ice slabs. The thresholds vary by only $\sim\pm$ 1 dB depending on the training region. The F1 score on the withheld portion of the data set (e.g. the measure of how well the threshold generalizes to the rest of the ice sheet) varies between 0.78 and 0.84. Given that the F1 score when using the entire dataset for training is 0.81, this suggests very good generalizability for thresholds derived from only a subset of the data.

Not surprisingly, there is more variation in the results for delineating the lower elevation limit of the ice slabs, with distinct region variations. In particular, thresholds derived only from data in regions NE and N1 do not generalize well to the rest of the ice sheet, whereas the rest of the training regions generalize well. This might be for several reasons. First, the northern regions have the least number of ice slab detections, so thresholds derived from data in those regions may be overfit to conditions that are not representative of larger areas. Second, we see steeper gradients in backscatter as a function of elevation in the northern regions compared to the Northwest and Southwest. This suggests that small variations in threshold values would lead to large changes in ice slab area in the Northwest and Southwest, but small changes in ice slab area in the North and Northeast. As a result, the agreement between the OIB observations and S-1 detections is much more sensitive to the threshold values in the Northwest and Southwest than in the North and Northeast, so thresholds derived from NE and N1 do not generalize well. Finally, these regional variations might represent how well the lower boundary of refrozen ice in each region actually agrees with the modeled long-term equilibrium line used to cutoff the OIB detections, and therefore reflect errors or uncertainties in that model.

[Figure]

**Figure 4.** Results of the 10-fold cross-validation scheme. The overview map of Greenland shows the ten training regions. In each panel, The training regions that produce the maximum and minimum total ice slab extent are marked in the grey bars. a) The total ice slab area and F1 score on the withheld validation set for each iteration of the ten-fold cross-validation of the ice slab upper elevation limit. b) The estimated values of $\alpha$ and $\beta$ derived from each of the ten training regions. c) The total ice slab area and F1 score on the withheld validation set for each iteration of the ten-fold cross-validation of the ice slab lower elevation limit. d) The estimated values of $\phi$ derived from each of the ten training regions.

**Minor comments**

**[R4]** *In paragraph 1 of the introduction please also mention that mass is also lost due to dynamical processes, and it would be helpful to briefly compare this mass loss to that from surface processes.*
We have edited the opening of the first paragraph to read: "Over the last two decades, around half of mass loss from the Greenland Ice Sheet (GrIS) has come from the runoff of surface meltwater, with the remaining 45-50% attributable to ice dynamical processes and ice-ocean interactions in marine terminating sectors (Van Den Broeke et al., 2009; Enderlin et al., 2014; Mouginot et al., 2019). However, surface processes are projected to remain the dominant contributor to Greenland's sea level contribution over the next century, particularly as the ice margin retreats onto land above sea level (Fox-Kemper et al., 2021). By extension, much of the uncertainty in future mass loss from the ice sheet can also be ascribed to uncertainty in surface processes (Fox-Kemper et al., 2021)."

**[R5]** *L28: "preferences the formation of perennial firn aquifers" is a bit awkward wording.*
We have edited this sentence (line 31) to read:
"The southeast basin is the only major region where no ice slabs have been detected, due to the high snow accumulation rate that insulates subsurface liquid water from refreezing and leads to the formation of perennial firn aquifers (Forster et al., 2014; Munneke et al., 2014)."

**[R6]** *L42: "... including the first high elevation rain event, such as 2019, 2021, and 2023." This wording makes it sound like the rain event occurred in 2019, 2021, and 2023.*
We have edited this sentence (line 45) to read:
"With the end of the OIB mission in 2019, there are no current or planned ice-penetrating radar

missions to improve these time series or to assess the impact of more recent heavy melt seasons, such as 2019, 2021, and 2023, which included a significant high elevation rain event in August 2021 (Tedesco and Fettweis, 2020; Box et al., 2022, 2023)."

**[R7]** *L111: From Fig. 1, it looks like the HV backscatter is closer to -4 dB in the percolation zone.* Corrected in the text as suggested.

**[R8]** *L112: "...eventually reaching a plateau around -11 dB". This is a bit misleading, I think. There is still substantial variation around this new plateau as the HV backscatter changes from -8 dB in the upper part of the ablation zone to -13 dB in the wet snow zone.*
We have edited this sentence (lines 118-121) to read:
"The percolation zone HV backscatter ($\sigma_{HV}^0$) is consistently about -4 dB, but decays at lower elevations as ice slabs begin to form and thicken, eventually plateauing around an average of -11 dB across the upper ablation and wet snow zones. However, there is significant local variability in the upper ablation and wet snow zones, with the HV backscatter varying from -13 dB to -6 dB around the mean."

**[R9]** *L191: "... we optimize independent backscatter thresholds..." What are the thresholds independent from?*
The backscatter thresholds are independent from one another – the choice of backscatter threshold to delineate the upper elevation limit of the ice slabs has no bearing on the threshold for the lower limit and vice versus (e.g. no joint optimization). To clarify this point, we have edited this sentence to read: "...we optimize **separate** backscatter thresholds...".

**[R10]** *L196: What is meant by "high-end estimate"?*
Jullien et al. (2023) developed a semi-automated routine for delineating ice slabs in ice-penetrating radar data, tuned to in-situ measurements from firn cores and GPR measurements at KAN-U the produced both a minimum and maximum likely ice presence. This is based on sensitivity tests that account for the fact that there is an overlap in the signal strength distributions between refrozen ice and porous firn. In their published data set accompanying their paper, they refer to the maximum likely ice presence estimate as the "high-end estimate" of ice slab extent, and we follow this terminology here. To clarify this point for readers who may be less familiar with their dataset, we have added the sentence:
"This high-end estimate corresponds to the maximum likely refrozen ice content given the observed ice-penetrating radar signal strength."

**[R11]** *L202: what step size did you use to test $\alpha$ and $\beta$ within these ranges?*
We used step sizes of 500 in digital number space for both $\alpha$ and $\beta$ (since data were original exported at 16-bit unsigned integers). In dB, this corresponds to a step size of 0.2 dB for $\alpha$ and 0.08 dB for $\beta$. Increasing the dB resolution for $\alpha$ to be the same at $\beta$ leads to no meaningful change in the results. The optimal of $\alpha$ is reduced by 0.04 dB leading to a 0.002 improvement in the F1 score (from 0.8114 to 0.8116). Similarly, when delineating the lower boundary, changing the dB resolution for $\gamma$ from 0.2 dB to 0.08 dB leads to a 0.08 dB change in $\gamma$ and no change in the F1 score.

**[R12]** *After the 10-fold validation, how were the optimal empirical parameters chosen?*
The optimal empirical parameters are based on the optimization using the entire OIB dataset from the whole ice sheet. We use the 10-fold validation to assess uncertainty in that optimal estimate by considering how the estimated ice slab extent might change if only part of that data were used for the

threshold optimization. From amongst the ten different sets of thresholds produced by this cross-validation scheme, we pick the two sets of thresholds that produce the largest and smallest total ice slab extent to conservatively represent this uncertainty range. In response to this comment, as well as comments from both R2 and R3, we have edited this portion of the manuscript describing the 10-fold cross-validation scheme to improve clarity (see the revised Section 3.3).

**[R13]** *L243: Please add Dunmire et al 2021 with Koenig et al 2015 citation.*
Added.

**[R14]** *Lines 285-290: I find this section confusing. Isn't "ice formed be refreezing" (L286) the same as an "ice slab"? The distinction between ice slabs and other refrozen ice is unclear throughout this section. Also, it seems that "ice formed by refreezing induces significant volume scattering due to trapped air bubbles..." (L286) contradicts the introduction "with relatively little volume scattering since heterogeneities such as air bubbles are significantly smaller than the C-band wavelength" (L95).*
We agree that this section can be somewhat confusing, in part because of imprecision in the language we have to describe ice sheet facies that form in the equilibrium zone. Ice slabs are generally understood to be multi-meter thick layer of refrozen ice that are perched over any otherwise porous and permeable relict firn layer. In this technical sense, areas where refrozen ice sits directly on top of meteoric ice would not be considered ice slabs. In some places, this ice might meet the definition of superimposed ice, which is ice that forms by refreezing within the annual snowpack on top of an otherwise solid ice column. However, in other places, this layer of refrozen ice over meteoric ice might form where water drained through crevasses into deep firn, completely filling it before refreezing, or where older ice slabs have exhumed through advection and ablation, or through other modes that are not yet well characterized. We have clarified this point in the revised manuscript by explicitly listing the definitions given above for readers who may not be familiar with technicalities (see lines 342-350).

Good point on L286! What we wanted to express here is that refrozen ice may contain remanent interstitial firn or other void space of sizes closer to the radar wavelength due to heterogeneous infiltration and refreezing, not air bubbles as might be found in ice formed via compaction. We will revise this sentence to read: "We hypothesize that any ice formed by refreezing induces significant volume scattering due to trapped interstitial firn pockets, void space, or other heterogeneities in density..."

**Technical corrections**
**[R15]** *L22: Please add "meltwater" before "retention and runoff"*
Adjusted in the text as suggested.

**[R16]** *L32: Please add "elevation" in "upper elevation limit" (also for L194).*
Adjusted in the text as suggested.

**[R17]** *L150: Delete "to" before "correction" at the start of this line.*
Adjusted in the text as suggested.

**Response to Reviewer #2**

*This paper presents an investigation of the potential for detecting ice slabs in the Greenland Ice Sheet using Sentinel-1 HH and HV C-band radar backscatter data. The paper is interesting and seems to show some promise for the method. The authors provide an appropriate degree of assessment indicating the regions of most uncertainty. I believe the paper can be published after some revisions, as follows:*

Thank you for the thoughtful review!

**[R1]** *Why was 500 m resolution used? The authors never discuss this. Finer resolution would be of interest. Was there a reason it was not pursued?*

Since we use a backscatter threshold method for detecting ice slabs, it is critical to our results that the mosaics primarily reflect spatial variations in backscatter due to surface properties, rather than speckle, look angle, or temporal variations. Multi-looking the data to 500 m resolution gives us the necessary speckle reduction and significantly improves the linear correlation between backscatter and incidence angle as a result, which is also critical to achieving a good incidence angle correction. We conducted a sensitivity test over a region of Southwest Greenland spanning from the divide to the coast and found that 500 m resolution is a good balance between high resolution and ensuring reasonably good linear correlation between the backscatter and incidence angle. These results are shown in the figure below.

[Figure]

**Figure 1.** Sensitivity test for the spatial resolution of backscatter mosaics. a) Box and whisker plots showing the distribution of Pearson Correlation Coefficients between $\sigma^0_{HH}$ and incidence angle over our study area as a function of resolution. Red crosses show outliers. b) Box and whisker plots showing the distribution of Pearson Correlation Coefficients between $\sigma^0_{HV}$ and incidence angle over our study area as a function of resolution. Correlation improves as resolution degrades, with 500 m offering a reasonable balance between speckle suppression and spatial resolution.

In the revised manuscript, we have discussed these points more directly (see lines 153-162) and include the above figure and discussion in the Supplementary Information.

**[R2]** *The authors discuss their reasoning for using only 1 year of data in the paper's conclusions. I can see their points, but the paper would be more impactful if a multi-year study were performed. I recommend at a minimum that the authors describe their reason for using only a single year earlier in the paper.*

We have moved this discussion to Section 3.1 when we first introduce the dataset (line 130). We do feel strongly that this is already a long paper (particularly once we add some of the sensitivity

analyses introduced during review) and that it makes the most sense to conduct a follow-up study using multiple years of data once this algorithm has been established and published.

**[R3]** *Bottom of p. 2: should say radiometer not radar data. 2nd paragraph of p. 3: dielectric misspelled*

Corrected in text.

**[R4]** *Could the authors provide more information on the residual errors after the angle correction is performed, e.g. a plot or two of the data before and after angle correction? I'm assuming there is a lot of residual here due to the highly variable topography, etc. which makes such effects also a potential source of error that could be included in later discussions.*

We have added the figures and discussion below to Supplementary Information discussing the residuals from the linear incidence angle correction. In general, we find that when we multilook to 500 m resolution before calculating the linear regression, the residuals are quite reasonable (mostly less than 2B – see panels a and b below). We also show in a detailed sensitivity analysis for Reviewer #1 (see that response for details) that the ice slab detection results are insensitive to the number of observations per pixel as long as it exceeds ~117 observations and insensitive to the range of incidence angles if there are measurements from at least 10 different incidence angles per pixel. When using the full data set from 2016-10-01 to 2017-04-30 as originally presented in the paper, these conditions are met for our area of interest.

[Figure]

**Figure 2.** Linear incidence angle correction residuals and slopes for both polarizations. a) Root mean square error (RMSE) of the linear fit between $\sigma^0_{HH}$ and incidence angle per pixel. b) RMSE of the linear fit between $\sigma^0_{HV}$ and incidence angle per pixel. c) Slope of the linear fit between $\sigma^0_{HH}$ and incidence angle per pixel. d) Slope of the linear fit between $\sigma^0_{HV}$ and incidence angle per pixel. White and red dots mark the six sites shown in Figure S3.

The regions with the largest residuals, such as firn aquifer areas on the east coast (site #4 in the above image) have large residuals because of large temporal variations in backscatter, rather than obvious failures to follow a linear trend. The figure below shows, on the left, scatter plots of incidence angle vs. backscatter for both polarizations and on the right, the time series of backscatter. The top two panels show a firn aquifer site (#4) compared with percolation zone site (#1) to illustrate the impact of time-varying backscatter. This makes the important point that even in regions with relatively stable winter backscatter time series, our incidence angle correction serves a second

purpose – it averages the backscatter in time to achieve an estimate of the mean winter backscatter that does not reflect small-scale temporal variations due to snowfall events, etc. In this way, some amount of residual expected and reasonable, since it quantifies the temporal variability in backscatter that we average out.

[Figure]

**Figure 3.** a) Scatterplot of backscatter vs. incidence angle at Site 1 in the percolation zone for both polarizations. b) Scatterplot of backscatter vs. time from 2016-10-01 to 2017-04-30 at Site 1. c) Scatterplot of backscatter vs. incidence angle at Site 2 in the ice slab zone for both polarizations. d) Scatterplot of backscatter vs. time from 2016-10-01 to 2017-04-30 at Site 2. e) Scatterplot of backscatter vs. incidence angle at Site 3 in the ablation zone for both polarizations. f) Scatterplot of backscatter vs. time from 2016-10-01 to 2017-04-30 at Site 3. g) Scatterplot of backscatter vs. incidence angle at Site 4 in a firn aquifer area for both polarizations. h) Scatterplot of backscatter vs. time from 2016-10-01 to 2017-04-30 at Site 4. i) Scatterplot of backscatter vs. incidence angle at Site 5 over a supraglacial lake for both polarizations. j) Scatterplot of backscatter vs. time from 2016-10-01 to 2017-04-30 at Site 5. k) Scatterplot of backscatter vs. incidence angle at Site 6 over a heavily crevassed region for both polarizations. l) Scatterplot of backscatter vs. time from 2016-10-01 to 2017-04-30 at Site 6.

**[R5]** *Figure 2 could be more appealing if zoomed somehow in the manner of Figure 6. As is, we mostly see the interior ice sheet regions that are not of interest in this study. Perhaps rotate 90 degrees and separate into rotated images of the East and West coasts of the ice sheet?*

We think that there is still value in showing the entire mosaic, in part because it emphasizes the non-uniqueness of the ice slab radiometric signature and the need to filter out the firn aquifer and dry snow zone regions. We also attempted to rearrange this figure as suggested, but the figure quickly becomes extremely large with many panels and the need for additional locator maps to explain where each panel was located on the ice sheet. Given that the full datasets will be published with the paper, we also think that the interested reader can easily download and zoom around on the full mosaics themselves. However, as a compromise, we have now included zoom-in panels over a selected region over the North Greenland ice slabs in Figure 2 to show a clearer example of spatial variations in backscatter in our regions of interest. The revised figure is shown below:

[Figure]

**[R6]** *Was the same angle correction used for the Summer data? If so, this should also be noted as a potential (minor) issue.*

Yes, the same incidence angle correction method is applied to the summer data. We will note this explicitly in the revised text.

**[R7]** *The apparent alignment of the contours in Figure 3 with a "45 degree line" suggests that a classifier based on sigma_HH + alpha sigma_HV where alpha is some constant might also be successful here, rather that the separate thresholds on each quantity?*

This certainly might work! However, we think we would still need to optimize two separate quantities – the $\alpha$ in the summation suggested above and some $\beta$ threshold for the summation image that delineates between ice slabs and not ice slabs. For this reason, it does not seem that it would significantly simplify the algorithm, so we did not pursue this method in revising this paper.

**[R8]** *Figure 5 appears somewhat redundant given that everything has been described in the text by this point.*
We think the visual presentation of the algorithm might still be helpful to some readers, but if the editor is concerned about the length of the paper, this will be the first thing we cut.

**[R9]** *I found the distinction between the "minimum", "maximum", etc. classifiers somewhat confusing. Please introduce these ideas more clearly early on in the discussion of the classifier.*
We have thoroughly revised Section 3.3 in the manuscript in response to this comment and suggestions from other reviewers. The revised text is shown below in the blue text, along with a revised Figure 4.

[revised manuscript text omitted]

**[R10]** *Dashed lines in Figures 6 and 11 are hard to discern especially when the topographic lines are included. Is there a way of doing this that makes them more clear? This comparison is key so making it easy to follow is crucial.*
In both Figure 6 and 11, we have changed the dashed lines to solid lines, which we believe makes the outlines clearer and easier to interpret. The new versions of the figures are shown below:

[Figure]

**Revised Figure 6.** Dashed lines in Panel A have been changed to solid line.

[Figure]

**Revised Figure 11.** All dashed lines have been changed to solid lines.

**[R11]** *Some of the discussions of differences seemed a little long and overly complex. Consider trying to simplify these discussions if possible, i.e. simply state "may have more volume scattering compared to xxx" etc.*
As we revised the paper, we have worked to make the discussion more concise.

**Response to Reviewer #3**

**General comments**
*The paper is interesting and introduces an empirical algorithm for detecting and monitoring the ice slab regions across the entire Greenland by using Sentinel-1 products. The text is well written and easy to understand. It opens by introducing the scientific problem, the physical mechanism upon which rely the algorithm (interaction between electromagnetic waves and the ice sheet), the regions excluded from the analysis to limit the noise in the problem, and the algorithm itself, along with the methodology for setting up the thresholds. Then it keep on with a suitable description of the results along with a fair discussion about these achievements and the uncertainties in the process. A comparison with previous mapping done with SMAP is also provided. I haven't found any major issue for the paper publication however I feel that some points should be improved before to proceed.*

Thank you for taking the time to review and provide constructive comments on our manuscript!

**Specific comments**
**[R1]** *The description of the "ten-fold cross-validation scheme" is quite convoluted and not straightforward to understand. It has to be improved.*
We have thoroughly revised Section 3.3 in the manuscript in response to this comment and suggestions from other reviewers. The new text is provided below in the blue font, along with an updated Figure 4.

[revised manuscript text omitted]

**Minor points**
**[R2]** *In several points of the text: I would refrain from the use of statements like "excellent agreement" and prefer something more mild as "fine agreement".*
We have softened these statements or reword them in response to both this review and comments from other the other reviewers. For example, in the abstract, we have made the following changes:

**[Original]** "The S-1 inferred ice slab extent is in excellent agreement with ice-penetrating radar ice slab detections from spring 2017."

**[Revised]** "Our results show that there is a sufficiently strong relation between C-band backscatter and the ice content of the upper ~7 meters of the firn column to enable ice slab mapping with S-1."

**[R3]** *lines 65-66: the sentence "...the depth-integrated surface echo measured by the instrument contains information about the near-surface structure" is usually true, however C-band SAR data can be affected also by phenomena originating deep in the ice. For instance subglacial Vostok Lake, Antarctica, clearly visible in the Radarsat image map of Antarctica. I would say something like "...contains information mainly about the near-surface structure".*

Good point – we have edited this sentence as suggested.

**[R4]** *line 78: delete the s at "cms". It is a SI symbol and doesn't require the s for the plural.*

Edited in text as suggested.

**[R5]** *Figure 1: the image of Greenland with the A-A' transect is very useful but must be better highlighted. As it is put, it gets unnoticed since the reader attention goes immediately to the top or bottom panel which are full of colors. Instead the map should be seen first. An option could be moving the top panel legend northwest, and then replace it with the Greenland map. Anyhow any different solution is fine.*

We have revised this figure as suggested and moved the map to the top panel. The revised figure is below:

[Figure]

**[R6]** *Line 120: what "Agency" means? Also at line 138.*

Thanks for noting this! This is an error in the display of a citation to an ESA technical document and has been fixed in the manuscript.

**[R7]** *Line 127: I notice that speckle filtering is not considered in the processing chain while it is a fundamental step for SAR processing at high resolution. Is there a justification for not applying it?*
We mitigate speckle through multi-looking in both space and time. Data are first multi-looked to 500 m resolution. In each pixel, we then estimate a linear relationship between incidence angle and backscatter based on data points from all images collected between 2016-10-01 to 2017-04-30. Finally, we calculate the theoretical average backscatter at 35-degree incidence angle from this relationship. This has the effect of averaging all backscatter values in a given pixel across that whole time period. Therefore, in each pixel in our mosaics, we have typically averaged together at least 200 separate measurements. The averaging in time is necessary not just for speckle reduction, but to smooth out temporal variations in backscatter due to random snowfall events, wind scouring, etc. However, we agree that in future work, it could be interesting to explore more sophisticated speckle filtering techniques to see if we could produce a robust map of ice slab extent at higher resolution

than 500 m. In our revised manuscript, we have included a clarifying discussion of this averaging approach when we discuss the resolution of the mosaics (see lines 153-162).

**[R8]** *Figure 2, caption, third line: "We excluded all the regions outside..." I suggest using a positive sentence as in the main body of the paper. "We considered only the regions..." is easier to understand.*
Edited in the text as suggested.

**[R9]** *Figure 2, caption: The disclaimer "Contains modified Copernicus Sentinel data 2016-2017, processed by ESA." makes the text heavy and is not informative at all. I suggest putting the disclaimer in the References and leave here a citation (or write a footnote). The same apply to Figure 1 caption.*
The ESA terms of use require this statement and our interpretation of the license is that it should accompany the relevant image. However, to de-emphasize this non-scientific information, we have moved it from the beginning to the end of the figure captions.

**[R10]** *Figure 3, top panel: the colormap of the image makes it difficult to be read given it compress the almost entire information in dark similar colors (F1>0.2). A different colormap should be used.*
Unfortunately, this is less an issue with the colormap, and more with the fact that many of the F1 values are with about 0.05 of the optimal F1 value. If we use the full range of the colorbar to show these values, then values from ~0-0.7 will be highly saturated instead. We have added a zoom-in panel to the region around the optimal threshold to better highlight the small changes in F1 value around the optimal point. The revised figure is shown below:

[Figure]

**Figure 3.** Selection of the optimal thresholds for ice slab detection. a) F1 score for delineating the upper elevation limit of the ice slabs as a function of $\alpha$ and $\beta$ thresholds. The optimal threshold combination (maximum F1 score) is shown in the white dot. b) Zoom-in of the region around the optimal threshold combination, showing the global maximum in F1 score. c) F1 score for delineating the lower elevation limit of the ice slabs as a function of $\phi$ threshold. The optimal threshold (maximum F1 score) is shown in the red dot.

**[R11]** *Line 332 and on: actually high-resolution data from PALSAR sensor (either onboard ALOS and ALOS-2) are already available without the need of waiting for NISAR or ROSE-L. Perhaps the issues are the full coverage of GrIS, the data of acquisition or the price of the products. I think the sentence can be better formulated.*

Thanks, this is a reasonable point. The main issue is indeed that ALOS-2 products are not freely available for scientific research and the ALOS-1 coverage of the GrIS is fairly patchy in most years. In particular, this makes it more difficult to average out the effects of speckle and temporal variations in backscatter to get a reliable mosaic, since there is typically only one acquisition over a given area per year, even in years with good coverage. However, we have amended this discussion to note that the ALOS-1 data is available and likely sufficient for an initial proof-of-concept testing of an L-band algorithm.

---

## Author Response (AR2)

**Response to Editor and Reviewer Comments**

**Editor Comments:**

**[1]** *Thank you for revising your manuscript thoroughly. The reviewers think your paper is much improved after revision. The third reviewer has a few minor comments which I think should be simple to address. I encourage you to take into account these comments in the revised paper.*

Thank you! We have revised the manuscript following the suggestions from Reviewer #3. In particular, we have clarified the minimum and maximum ice slab extent and the regions where the backscatter mosaics meet the minimum thresholds for number of observations and angular diversity per pixel.

**Reviewer #1 Comments:**

**[1]** *Authors have done a good job responding to the review comments.*
Thank you!

**Reviewer #2 Comments:**

**[1]** *The Authors replied exhaustively to all of my comments and suggestions. To me the paper is ready to be published.*
Thank you!

**Reviewer #3 Comments:**

**[1]** *I appreciate all the work that the authors put into the new version of the manuscript and find the new additions to the work helpful and interesting! I only have a few minor comments to consider.*
Thank you!

**[2]** *The new addition of regional 10-fold cross-validation is very helpful. However, I am still left wondering what impact this has on ice slab extent? In L323, the authors list the most likely ice slab extent but I don't believe they ever mention the minimum or maximum extents. I think a few sentences addressing this range should be added to the results or discussion.*
Thanks for catching this omission! We have added the total area from the minimum and maximum ice slab extents to the same sentence where we give the most likely ice slab extent. We have also added a short discussion about the differences. We particularly note that most of the large differences between the most likely and minimum ice slab total area are drive by large uncertainties in the lower elevation limit of the ice slabs, while the upper elevation limit agrees quite well across all estimates. See lines 330-340 in the revised manuscript for the full discussion.

**[3]** *L68 – EESA to ESA*
Corrected in the text.

**[4]** *L114 - $\sigma_{xpol}$ is mentioned before it is defined (L121)*
We have reworked the flow of these two paragraphs to introduce the definition of $\sigma_{xpol}$ when it is first mentioned at L114. See lines 114-217 in the revised manuscript.

**[5]** *L162 – A word Is needed between "fit" and "the" in this sentence.*
Corrected in the text.

**[6]** *L167-169: "The final ice slab classification results are insensitive to angular diversity or number of observations as long as there are a median of at least ~117 observations per pixel spanning at least 10 unique incidence angles, a criterion which is met for our study area." This is not exactly true since in L147, the authors say that the minimum pixel observations was 29. What proportion of the study area has >117 observations?*
Yes, we were imprecise in our wording here. What we intended to convey is that these thresholds are met in our regions of interest, e.g. in the areas where ice slabs are found, since the low observation regions are in the interior of the ice sheet and are filtered out by the dry snow zone mask before threshold optimization. It is also worth noting that the 117 observations can be relaxed to 77 observations if the median incidence angle exceeds 30 degrees, which is the case for our dataset in 2016-2017. In the supplementary information at lines 53-61, we now describe this relaxed threshold and have added a new Figure S5 that shows the region of the ice sheet where the observation and angular diversity criteria are met relative to where ice slabs are found. In the main text, we are now careful to say that these criteria are met in areas with ice slabs and provide pointers to the supplementary discussion and figure show the exact regions (see lines 169-175).

**[7]** *L268 – 'teh' typo*
Corrected in the text.